# Tumor elimination by clustered microRNAs miR-306 and miR-79 via noncanonical activation of JNK signaling

Zhaowei Wang[1,2], Xiaoling Xia[2,3], Jiaqi Li[2], Tatsushi Igaki[2]*

[1]State Key Laboratory of Biocontrol, School of Ecology, Sun Yat-sen University, Shenzhen, China; [2]Laboratory of Genetics, Graduate School of Biostudies, Kyoto University, Yoshida-Konoe-cho, Kyoto, Japan; [3]Guangzhou Key Laboratory of Insect Development Regulation and Application Research, Institute of Insect Science and Technology & School of Life Sciences, South China Normal University, Guangzhou, China

*For correspondence:
igaki.tatsushi.4s@kyoto-u.ac.jp

Competing interest: The authors declare that no competing interests exist.

**Abstract** JNK signaling plays a critical role in both tumor promotion and tumor suppression. Here, we identified clustered microRNAs (miRNAs) miR-306 and miR-79 as novel tumor-suppressor miRNAs that specifically eliminate JNK-activated tumors in *Drosophila*. While showing only a slight effect on normal tissue growth, miR-306 and miR-79 strongly suppressed growth of multiple tumor models, including malignant tumors caused by Ras activation and cell polarity defects. Mechanistically, these miRNAs commonly target the mRNA of an E3 ubiquitin ligase ring finger protein 146 (RNF146). We found that RNF146 promotes degradation of tankyrase (Tnks), an ADP-ribose polymerase that promotes JNK activation in a noncanonical manner. Thus, downregulation of RNF146 by miR-306 and miR-79 leads to hyper-enhancement of JNK activation. Our data show that, while JNK activity is essential for tumor growth, elevation of miR-306 or miR-79 overactivate JNK signaling to the lethal level via noncanonical JNK pathway and thus eliminate tumors, providing a new miRNA-based strategy against cancer.

## Editor's evaluation

This article is valuable as it uncovers a previously unknown tumor-suppressor mechanism that eliminates *JNK*-activated *Drosophila* tumors. This mechanism is triggered by the overexpression of microRNAs that downregulate an E3 ubiquitin ligase RNF146, whose loss causes an increase in Tnks (poly-ADP-ribose polymerases) and *JNK* signaling. This tumor-suppressor mechanism has potential implications for the treatment of *JNK*-activated tumors. This article is of interest to people in the tumor suppressor, *JNK*, and miRNA fields, and the key claims are convincing and well supported by the data, and the authors use thoughtful and rigorous approaches.

## Introduction

Cancer progression is driven by oncogenic alterations of intracellular signaling that lead to promotion of cell proliferation and suppression of cell death (***Croce, 2008***). The c-Jun N-terminal kinase (JNK) pathway is an evolutionarily conserved mitogen-activated protein (MAP) kinase cascade that regulates both cell proliferation and cell death in normal development and cancer (***Bode and Dong, 2007***; ***Eferl and Wagner, 2003***). Indeed, JNK signaling can act as both tumor promoter and tumor suppressor depending on the cellular contexts (***Bode and Dong, 2007***; ***Bubici and Papa, 2014***; ***Karin and Gallagher, 2005***). Crucially, JNK signaling is often activated in various types of cancers (***Bubici and Papa,***

2014; *Wu et al., 2019*). Thus, accumulating evidence suggests that JNK signaling can be a critical therapeutic target for cancer. For instance, converting JNK's role from pro-tumor to antitumor within tumor tissue could be an ideal anticancer strategy.

*Drosophila* provides a superb model for studying the genetic pathway of cellular signaling and has made great contributions to understand the basic principle of tumor growth and progression (*Enomoto et al., 2018*; *Tipping and Perrimon, 2014*). The best-studied model of *Drosophila* malignant tumor is generated by clones of cells overexpressing oncogenic Ras (Ras$^{V12}$) with simultaneous mutations in apicobasal polarity genes such as *lethal giant larvae* (*lgl*), *scribble* (*scrib*), or *discs large* (*dlg*) in the imaginal epithelium (*Brumby and Richardson, 2003*; *Pagliarini and Xu, 2003*). These tumors activate JNK signaling and blocking JNK within the clones strongly suppresses their tumor growth (*Igaki et al., 2006*; *Uhlirova and Bohmann, 2006*), indicating that JNK acts as a pro-tumor signaling in these malignant tumors. Conversely, clones of cells overexpressing the oncogene Src in the imaginal discs activate JNK signaling and blocking JNK in these clones results in an enhanced overgrowth (*Enomoto and Igaki, 2013*), indicating that JNK negatively regulates Src-induced tumor growth. Similarly, although clones of cells mutant for *scrib* or *dlg* in the imaginal discs are eliminated by apoptosis when surrounded by wild-type cells, blocking JNK in these clones suppresses elimination and causes tumorous overgrowth (*Brumby and Richardson, 2003*; *Igaki et al., 2009*), indicating that JNK acts as antitumor signaling in these mutant clones. Thus, JNK also acts as both pro- and antitumor signaling depending on the cellular contexts in *Drosophila* imaginal epithelium.

miRNAs are a group of small noncoding RNAs that suppress target gene expression by mRNA degradation or translational repression and have been proposed to be potent targets for cancer therapy. Indeed, several cancer-targeted miRNA drugs have entered clinical trials in recent years. For instance, MRX34, a miRNA mimic drug developed from the tumor-suppressor miR-34a, is the first miRNA-based anticancer drug that has entered phase I clinical trials for patients with advanced solid tumors (*Beg et al., 2017*; *Hong et al., 2020*). In addition, MesomiR-1, a miR-16 mimic miRNA that targets EGFR, has entered phase I trial for the treatment of thoracic cancers (*Reid et al., 2013*; *van Zandwijk et al., 2017*). Such miRNA-mediated anticancer strategy can be studied using the *Drosophila* tumor models. Indeed, in *Drosophila*, the conserved miRNA let-7 targets a transcription factor *chinmo* and thus suppresses tumor growth caused by *polyhomeotic* mutations (*Jiang et al., 2018*). In addition, miR-8 acts as a tumor suppressor against Notch-induced *Drosophila* tumors by directly inhibiting the Notch ligand Serrate (*Vallejo et al., 2011*). However, apart from these miRNAs that suppress growth of specific types of tumors, it is unclear whether there exist miRNAs that generally suppress tumor growth caused by different genetic alterations.

Here, using *Drosophila* tumor models and subsequent genetic analyses, we identified several tumor-suppressor miRNAs. Among these, miR-306 and miR-79, two clustered miRNAs located on the miR-9c/306/79/9b cluster, significantly suppressed growth of multiple types of JNK-activated tumors while showing only a slight effect on normal tissue growth. Mechanistically, miR-306 and miR-79 directly target *RNF146*, an E3 ubiquitin ligase that causes degradation of a JNK-promoting ADP-ribose polymerase Tnks, thereby overamplifying JNK signaling in tumors to the lethal levels via noncanonical JNK activation. Our findings provide a novel miRNA-based strategy that generally suppress growth of JNK-activating tumors.

## Results

### Identification of miR-306 and miR-79 as novel tumor-suppressor miRNAs

To identify novel antitumor miRNAs in *Drosophila*, we focused on 37 miRNA clusters or miRNAs that are highly expressed in *Drosophila* eye-antennal discs (*Chung et al., 2008*). Using the Flippase (FLP)-Flp recognition target (FRT)-mediated genetic mosaic technique, each miRNA was overexpressed in clones of cells expressing Ras$^{V12}$ with simultaneous mutations in the apicobasal polarity gene *dlg* (Ras$^{V12}$/*dlg*$^{-/-}$) in the eye-antennal discs, the best-studied malignant tumor model in *Drosophila* (*Pagliarini and Xu, 2003*; *Figure 1C*; compare to *Figure 1*). We found that overexpression of miR-7, miR-79, miR-252, miR-276a, miR-276b, miR-282, miR-306, miR-310, miR-317, miR-981, miR-988, or the miR-9c/306/79/9b cluster in Ras$^{V12}$/*dlg*$^{-/-}$ clones dramatically suppressed tumor growth (*Figure 1C–G, Figure 1—figure supplement 1D, R, S, T, W, Y, Z, AC, and AE*, quantified in *Figure 1H*

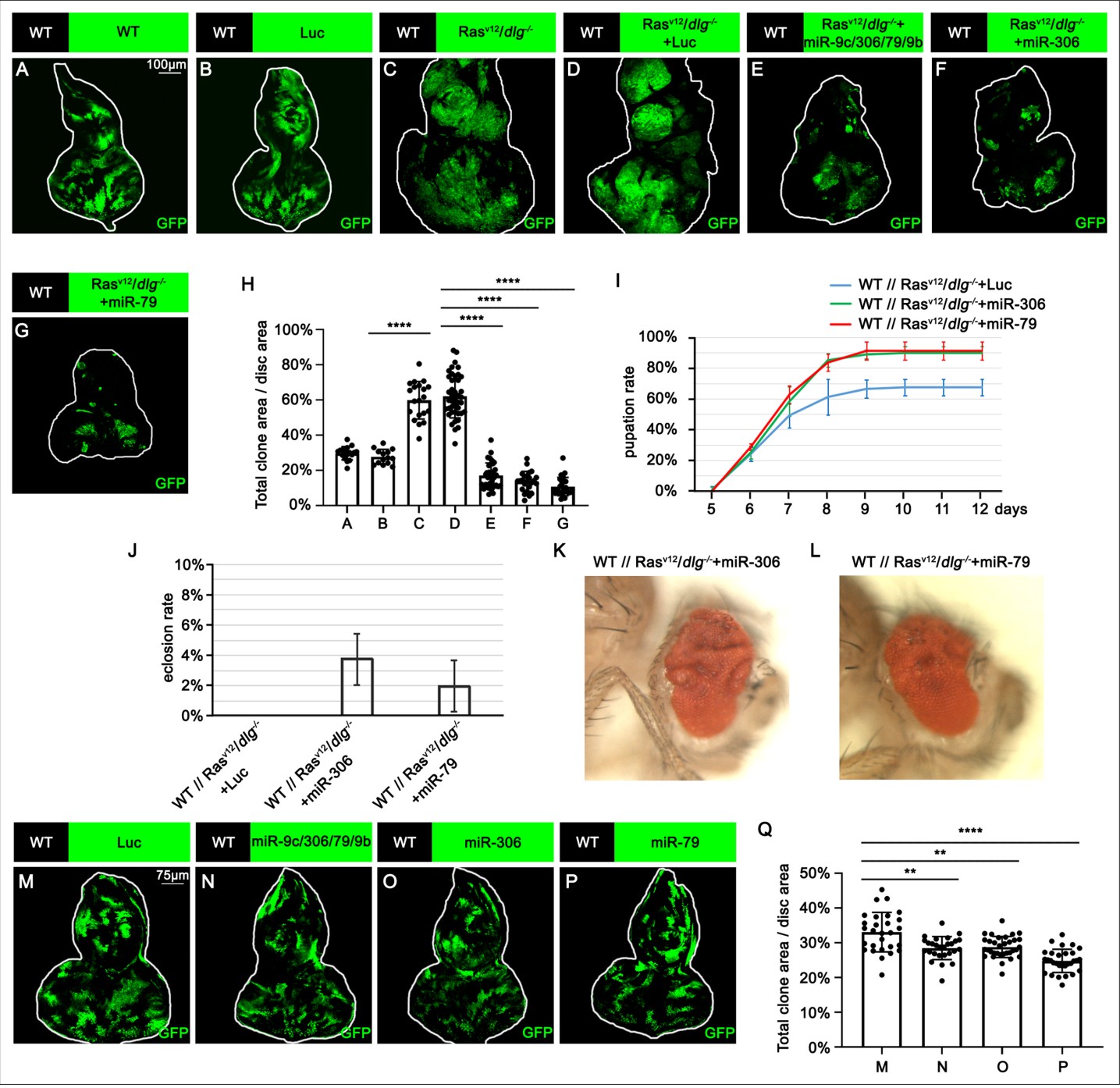

**Figure 1.** miR-306 and miR-79 suppress Ras[v12]/*dlg*[-/-] tumor growth. (**A–G**) Eye-antennal disc bearing GFP-labeled clones of indicated genotypes (**A** and **B**, 5 days after egg laying, **C–G**, 7 days after egg laying). (**H**) Quantification of clone size (% of total clone area per disc area in eye-antennal disc) in (**A–G**). Error bars, SD; ****p<0.0001 by one-way ANOVA multiple-comparison test. (**I**) Pupation rate of flies with indicated genotypes. Data from three independent experiment, n > 30 for each group in one experiment; error bars, SD. (**J**) Eclosion rate of flies with indicated genotypes. Data from three independent experiment, n > 30 for each group in one experiment; error bars, SD. (**K, L**) Adult eye phenotype of flies with indicated genotypes. (**M–P**) Eye-antennal disc bearing GFP-labeled clones of indicated genotypes (5 days after egg laying). (**Q**) Quantification of clone size (% of total clone area per disc area in eye-antennal disc) of (**M–P**). Error bars, SD; **p<0.01, ****,p<0.0001 by one-way ANOVA multiple-comparison test.

The online version of this article includes the following source data and figure supplement(s) for figure 1:

**Source data 1.** Quantitative data for *Figure 1*.

**Source data 2.** Genotypes for *Figure 1* and *Figure 1—figure supplements 1 and 2*.

*Figure 1 continued on next page*

*Figure 1 continued*

**Figure supplement 1.** Effect of miRNAs or miRNA clusters on Ras$^{V12}$/*dlg*$^{-/-}$ tumor growth.

**Figure supplement 1—source data 1.** Quantitative data for *Figure 1—figure supplement 1*.

**Figure supplement 2.** miR-306 and miR-79 suppress Ras$^{V12}$/*lgl*$^{-/-}$ tumor growth.

**Figure supplement 2—source data 1.** Quantitative data for *Figure 1—figure supplement 2*.

and *Figure 1—figure supplement 1AI*). In addition, overexpression of miR-305, miR-995, or the miR-13a/13b-1/2c cluster mildly suppressed Ras$^{V12}$/*dlg*$^{-/-}$ tumor growth (*Figure 1—figure supplement 1K, X and AF*, quantified in *Figure 1—figure supplement 1AI*). Clustered miRNAs are localized close to each other in the genome and are thus normally transcribed together, ensuring the transcription efficiency of miRNA genes (*Kabekkodu et al., 2018*; *Ryazansky et al., 2011*). Notably, overexpression of the miR-9c/306/79/9b cluster, miR-306, or miR-79 dramatically inhibited Ras$^{V12}$/*dlg*$^{-/-}$ tumor growth (*Figure 1E–G*, compare to *Figure 1D*, quantified in *Figure 1H*). In addition, overexpression of miR-306 or miR-79 was sufficient to rescue the reduced pupation rate and animal lethality caused by Ras$^{V12}$/*dlg*$^{-/-}$ tumors in the eye-antennal discs (*Figure 1I and L*). A pervious study in *Drosophila* wing discs showed that overexpression of miR-79 suppressed tumor growth caused by coexpression of Ras$^{V12}$ and *lgl*-RNAi via unknown mechanisms (*Shu et al., 2017*). Similarly, we found that overexpression of the miR-9c/306/79/9b cluster, miR-306, or miR-79 strongly suppressed growth of Ras$^{V12}$/*lgl*$^{-/-}$ tumors in the eye-antennal discs (*Figure 1—figure supplement 2A–D*, quantified in *Figure 1—figure supplement 2E*). Importantly, overexpression of the miR-9c/306/79/9b cluster, miR-306, or miR-79 alone only slightly reduced clone size compared to wild-type (*Figure 1M–P*, quantified in *Figure 1Q*). These data indicate that miR-306 and miR-79 are tumor-suppressor miRNAs that only mildly suppress normal tissue growth but specifically block tumor growth in *Drosophila* imaginal epithelium.

## miR-306 and miR-79 suppress tumor growth by promoting cell death

We next investigated the mechanism by which miR-306 and miR-79 suppress tumor growth. Immunostaining of Ras$^{V12}$/*dlg*$^{-/-}$ or Ras$^{V12}$/*lgl*$^{-/-}$ tumors with anti-cleaved DCP-1 antibody revealed that expression of the miR-9c/306/79/9b cluster, miR-306, or miR-79 in tumor clones significantly increased the number of dying cells (*Figure 2A–E*, *Figure 2—figure supplement 1A–D*, quantified in *Figure 2F* and *Figure 2—figure supplement 1E*). In addition, blocking cell death in tumor clones by overexpressing the caspase inhibitor baculovirus p35 canceled the tumor-suppressive activity of miR-306 or miR-79, while p35 overexpression alone did not affect growth of normal tissues or Ras$^{V12}$/*dlg*$^{-/-}$ tumors (*Figure 2G–N*, quantified in *Figure 2O*). These data indicate that the miR-9c/306/79/9b cluster, miR-306, or miR-79 suppresses tumor growth by inducing cell death. Importantly, overexpression of these miRNAs alone did not cause cell death in normal tissue (*Figure 2P–S*, quantified in *Figure 2T*), suggesting that miR-306 or miR-79 cooperates with a putative tumor-specific signaling activated in Ras$^{V12}$/*dlg*$^{-/-}$ or Ras$^{V12}$/*lgl*$^{-/-}$ tumors to induce synthetic lethality.

## miR-306 and miR-79 suppress tumor growth by enhancing JNK signaling

We thus examined whether Ras activation or cell polarity defect cooperates with miR-306 or miR-79 to induce cell death. Overexpression of the miR-9c/306/79/9b cluster, miR-306, or miR-79 in Ras$^{V12}$-expresing clones did not affect their growth (*Figure 3—figure supplement 1A–D*, quantified in *Figure 3—figure supplement 1E*), indicating that Ras signaling does not cooperate with these miRNAs. Notably, overexpression of these miRNAs in *dlg*$^{-/-}$ clones significantly reduced their clone size (*Figure 3A–E*, quantified in *Figure 3F*). In addition, blocking cell death by overexpression of p35 canceled the ability of these miRNAs to reduce *dlg*$^{-/-}$ clone size (*Figure 3G–O*, quantified in *Figure 3P*), suggesting that these miRNAs block *dlg*$^{-/-}$ clone growth by promoting cell death. These data show that miR-306 or miR-79 cooperates with loss of cell polarity to induce synthetic lethality.

We then sought to identify the polarity defect-induced intracellular signaling that cooperates with miR-306 or miR-79 to induce cell death. It has been shown that clones of cells mutant for cell polarity genes such as *dlg* activate JNK signaling via the *Drosophila* tumor necrosis factor (TNF) Eiger (*Brumby and Richardson, 2003*; *Igaki et al., 2009*). We found that overexpression of miR-306 or miR-79 alone moderately activated JNK signaling in the eye-antennal discs, as visualized by anti-p-JNK antibody

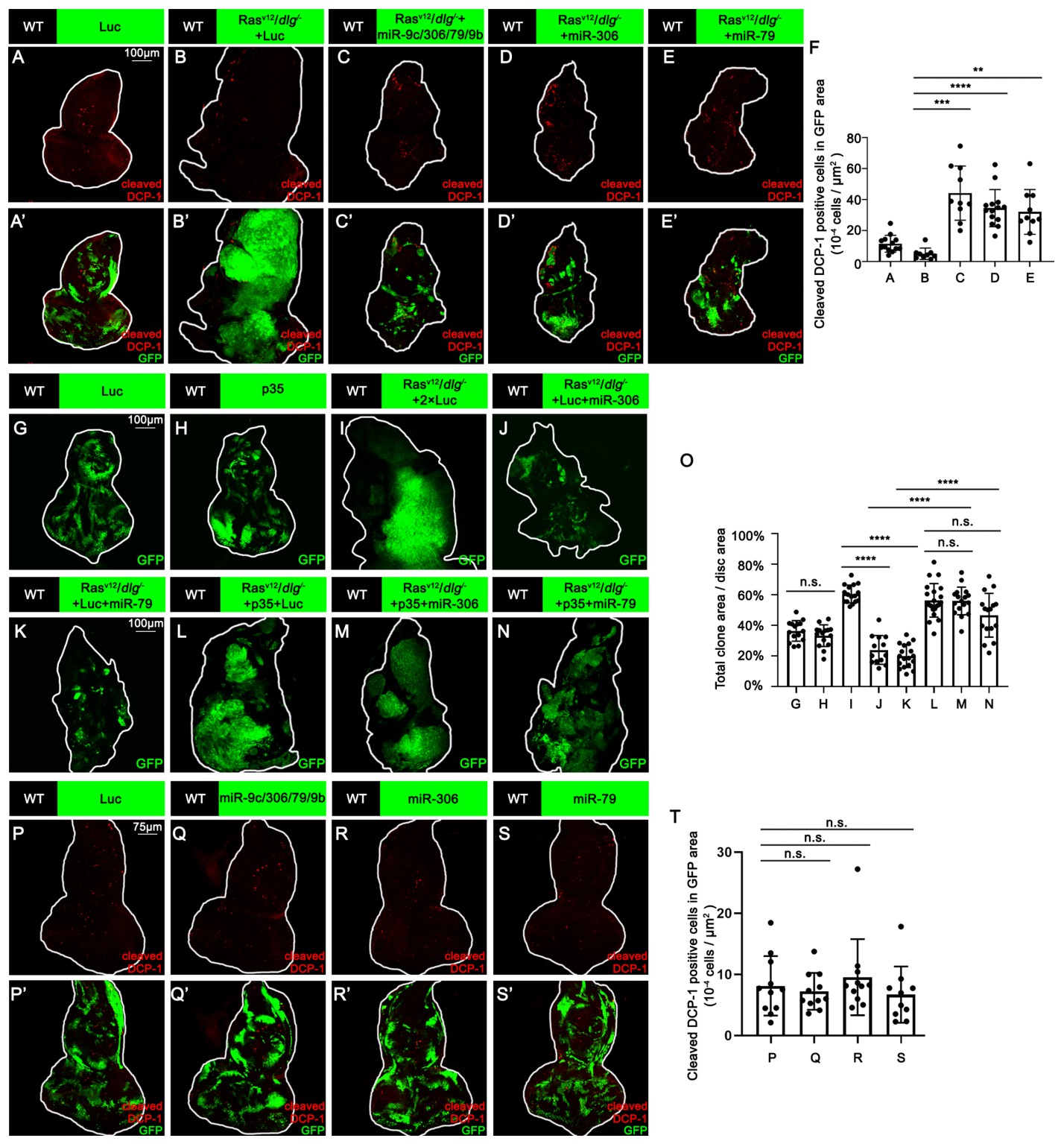

**Figure 2.** miR-306 and mir-79 suppress Ras$^{V12}$/*dlg*$^{-/-}$ tumor growth by inducing apoptosis. (**A–E**) Eye-antennal disc bearing GFP-labeled clones (A'-E') of indicated genotypes stained with anti-cleaved Dcp-1 antibody (A-E and A'-E', **A**, 5 days after egg laying, **B–E**, 7 days after egg laying). (**F**) Quantification of dying cells in GFP-positive clone area in (**A–E**). Error bars, SD; **p<0.01, ***p<0.001, ****p<0.0001 by one-way ANOVA multiple-comparison test. (**G–N**) Eye-antennal disc bearing GFP-labeled clones of indicated genotypes (**G** and **H**, 5 days after egg laying, **I–N**, 7 days after egg laying). (**O**) Quantification of clone size (% of total clone area per disc area in eye-antennal disc) of (**G–N**). Error bars, SD; n.s., p>0.05 (not significant), ****p<0.0001 by one-way ANOVA multiple-comparison test. (**P–S**) Eye-antennal disc bearing GFP-labeled clones (**P'-S'**) of indicated genotypes stained

*Figure 2 continued on next page*

*Figure 2 continued*

with anti-cleaved Dcp-1 antibody (P-S and P'-S', 5 days after egg laying). (**T**) Quantification of dying cells in GFP-positive clone area in (**P–S**). Error bars, SD; n.s., p>0.05 (not significant) by one-way ANOVA multiple-comparison test.

The online version of this article includes the following source data and figure supplement(s) for figure 2:

**Source data 1.** Quantitative data for *Figure 2*.

**Source data 2.** Genotypes for *Figure 2* and *Figure 2—figure supplement 1*.

**Figure supplement 1.** miR-306 and miR-79 induce apoptosis in Ras$^{V12}$/*lgl*$^{-/-}$ tumors.

**Figure supplement 1—source data 1.** Quantitative data for *Figure 2—figure supplement 1*.

staining and the *puc-LacZ* reporter (*Figure 3—figure supplement 2A–F*). In addition, Western blot analysis with anti-p-JNK antibody revealed that overexpression of miR-306 or miR-79 in the eyes using the GMR-Gal4 driver caused JNK activation (*Figure 3—figure supplement 2G*, quantified in *Figure 3—figure supplement 2H*). Notably, although overexpression of miR-306 or miR-79 alone in the eyes had no significant effect on eye morphology (*Figure 3—figure supplement 2I–K*, quantified in *Figure 3—figure supplement 2L*), they dramatically enhanced the reduced-eye phenotype caused by overexpression of Eiger (*Figure 3—figure supplement 2M–O*, quantified in *Figure 3—figure supplement 2P*). It has been shown that the severity of the reduced-eye phenotype depends on the levels of JNK activation and subsequent cell death (*Igaki et al., 2002*; *Igaki et al., 2006*; *Igaki et al., 2009*), suggesting that miR-306 and miR-79 enhance Eiger-mediated activation of JNK signaling. Indeed, blocking JNK signaling by overexpression of a dominant-negative form of *Drosophila* JNK Basket (Bsk$^{DN}$) canceled the tumor-suppressive activity of miR-306 or miR-79 against Ras$^{V12}$/*dlg*$^{-/-}$ or Ras$^{V12}$/*lgl*$^{-/-}$ tumors (*Figure 3R–W*, quantified in *Figure 3X*, and *Figure 3—figure supplement 3A–C*, quantified in *Figure 3—figure supplement 3D*), while Bsk$^{DN}$ did not affect growth of normal tissues (*Figure 3Q* compare to *Figure 3A*, quantified in *Figure 3X*). Moreover, overexpression of Bsk$^{DN}$ significantly increased the size of *dlg*$^{-/-}$ or *lgl*$^{-/-}$ clones overexpressing miR-306 or miR-79 (*Figure 3H, J, K, Y, Z, and AA*, quantified in *Figure 3AB*; *Figure 3—figure supplement 3E–J*, quantified in *Figure 3—figure supplement 3K*). Together, these data suggest that miR-306 and miR-79 suppress growth of malignant tumors by enhancing JNK signaling activation.

## miR-306 and miR-79 enhance JNK signaling stimulated by different upstream signaling

We next examined whether miR-306 or miR-79 suppresses growth of other types of tumors with elevated JNK signaling via an Eiger-independent mechanism. Overexpression of an activated form of the *Drosophila* PDGF/VEGF receptor homolog (PVR$^{act}$) results in JNK activation and tumor formation in the wing disc (*Wang et al., 2016a*) and eye-antennal disc (*Figure 4B*, compare to *Figure 4A*, quantified in *Figure 4F*). This tumor growth was significantly suppressed by overexpression of the miR-9c/306/79/9b cluster, miR-306, or miR-79 (*Figure 4B–E*, quantified in *Figure 4F*). In addition, the size of clones overexpressing the oncogene Src64B in the eye-antennal disc (*Figure 4H*, compare to *Figure 4G*, quantified in *Figure 4F*), which activate JNK signaling (*Enomoto and Igaki, 2013*), was significantly reduced when the miR-9c/306/79/9b cluster, miR-306, or miR-79 was coexpressed (*Figure 4H–K*, quantified in *Figure 4L*). Moreover, nonautonomous overgrowth of surrounding wild-type tissue by Src64B-overexpressing clones (*Enomoto and Igaki, 2013*) was significantly suppressed by coexpression of these miRNAs (*Figure 4M–P*, quantified in *Figure 4Q*). Furthermore, the size of clones mutant for an RNA helicase Hel25E or an adaptor protein Mahj, both of which are eliminated by JNK-dependent cell competition when surrounded by wild-type cells (*Nagata et al., 2019*; *Tamori et al., 2010*), was significantly reduced when these miRNAs were coexpressed (*Figure 4—figure supplement 1A–D*, quantified in *Figure 4—figure supplement 1E*, and *Figure 4—figure supplement 1F–I*, quantified in *Figure 4—figure supplement 1J*). In these tumors or cell competition models, ectopic expression of miR-306 or miR-79 enhanced JNK activity (*Figure 4—figure supplement 2*). We further examined whether expression of these miRNAs enhances normally occurring JNK activity during development. The pnr-GAL4 driver strain specifically expresses GAL4 in the wing discs in a broad domain corresponding to the central presumptive notum during metamorphosis (*Ishimaru et al., 2004*; *Zeitlinger and Bohmann, 1999*). Knocking down Hep, the *Drosophila* JNK kinase, using the pnr-GAL4 driver generates a split-thorax phenotype caused by reduced JNK signaling (*Ishimaru*

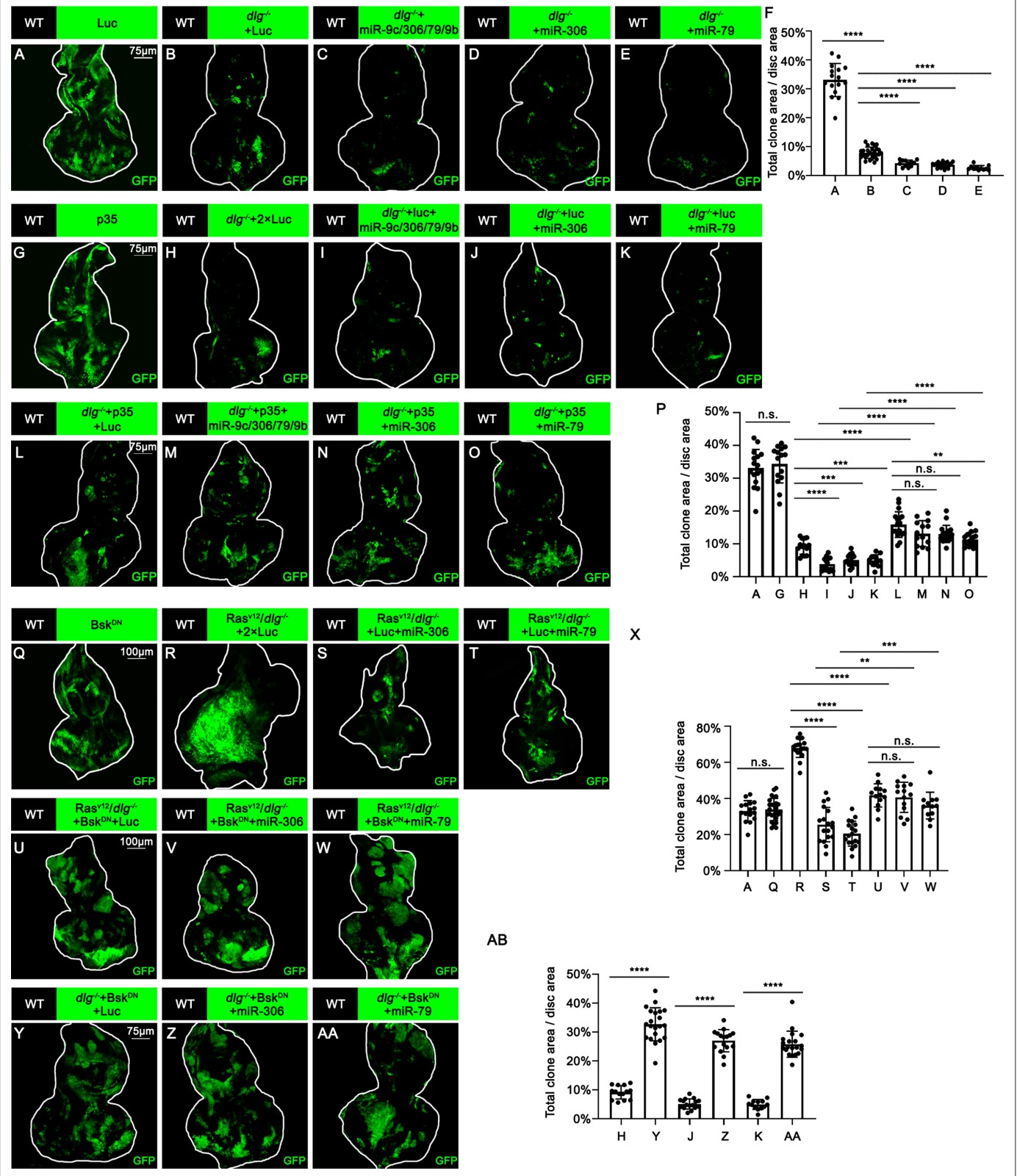

**Figure 3.** miR-306 and mir-79 suppress tumor growth and promote cell competition by promoting JNK signaling. (**A–E**) Eye-antennal disc bearing GFP-labeled clones of indicated genotypes (5 days after egg laying). (**F**) Quantification of clone size (% of total clone area per disc area in eye-antennal disc) of (**A–E**). Error bars, SD; ****p<0.0001 by one-way ANOVA multiple-comparison test. (**G–O**) Eye-antennal disc bearing GFP-labeled clones of indicated genotypes (5 days after egg laying). (**P**) Quantification of clone size (% of total clone area per disc area in eye-antennal disc) of (**A, G–O**). Error bars, SD;

*Figure 3 continued on next page*

*Figure 3 continued*

n.s., p>0.05 (not significant), **p<0.01, ***p<0.001, ****p<0.0001 by one-way ANOVA multiple-comparison test. (**Q–W**) Eye-antennal disc bearing GFP-labeled clones of indicated genotypes (**Q**, 5 days after egg laying, **R–W**, 7 days after egg laying). (**X**) Quantification of clone size (% of total clone area per disc area in eye-antennal disc) of (**A, Q–W**). Error bars, SD; n.s., p>0.05 (not significant), **p<0.01, ***p<0.001, ****p<0.0001 by one-way ANOVA multiple-comparison test. (**Y–AA**) Eye-antennal disc bearing GFP-labeled clones of indicated genotypes (5 days after egg laying). (**AB**) Quantification of clone size (% of total clone area per disc area in eye-antennal disc) of (**H, J, K, Y–AA**). Error bars, SD; ****p<0.0001 by one-way ANOVA multiple-comparison test.

The online version of this article includes the following source data and figure supplement(s) for figure 3:

**Source data 1.** Quantitative data for *Figure 3*.

**Source data 2.** Genotypes for *Figure 3* and *Figure 3—figure supplements 1–3*.

**Figure supplement 1.** miR-306 and miR-79 do not suppresses Ras$^{V12}$ tumor growth.

**Figure supplement 1—source data 1.** Quantitative data for *Figure 3—figure supplement 1*.

**Figure supplement 2.** miR-306 and miR-79 promote JNK signaling in the eye-antennal disc and adult eye.

**Figure supplement 2—source data 1.** Quantitative data or raw data for *Figure 3—figure supplement 2*.

**Figure supplement 3.** miR-306 and miR-79 suppress Ras$^{V12}$/*lgl*$^{-/-}$ tumor growth by promoting JNK signaling.

**Figure supplement 3—source data 1.** Quantitative data for *Figure 3—figure supplement 3*.

*et al., 2004*). On the contrary, ectopic expression of Hep or Eiger using pnr-GAL4 generates a small-scutellum phenotype caused by elevated JNK signaling (*Ma et al., 2013*; *Xue et al., 2007*). Similarly to Hep or Eiger, ectopic expression of miR-306 or miR-79 using pnr-GAL4 resulted in a small-scutellum phenotype (*Figure 4—figure supplement 3A–C*, quantified in *Figure 4—figure supplement 3D*). These data suggest that miR-306 and miR-79 broadly enhance JNK signaling activity stimulated by different upstream signaling.

## miR-306 and miR-79 enhance JNK signaling activity by targeting RNF146

We next sought to identify the mechanism by which miR-306 and miR-79 enhance JNK signaling by searching for the target gene(s) of these miRNAs. The clustered miRNAs often target overlapping sets of genes and thus co-regulate various biological processes (*Kim et al., 2009*; *Wang et al., 2016b*; *Yuan et al., 2009*). Given that miR-306 and miR-79 are located on the same miRNA cluster, we searched for the common targets of these miRNAs using the online software TargeyScanFly 7.2 (http://www.targetscan.org/fly_72/) and found 11 mRNAs that were predicted to be targets of both miR-306 and miR-79 (*Figure 5A*). We then examined whether knocking down of each one of these candidate genes could activate JNK signaling in *Drosophila* wing discs, where a clear JNK activation was observed when miR-306 or miR-79 was overexpressed (*Figure 5—figure supplement 1*). As a result, we found that knocking down of RNF146, but not any other available RNAis for the candidate genes, resulted in JNK activation (*Figure 5B and C*, *Figure 5—figure supplement 2*). The RNF146 mRNA had putative target sites of miR-306 and miR-79 in its 3′UTR region (*Figure 5D*). To confirm that RNF146 mRNA is a direct target of miR-306 and miR-79, we performed a dual-luciferase reporter assay in *Drosophila* S2 cells using wild-type RNF146 3′UTR (RNF146 WT) or mutant RNF146 3′UTR bearing mutations at the putative binding site of miR-306 (RNF146 m1) or miR-79 (RNF146 m2) (*Figure 5D*). We found that miR-306 and miR-79 reduced wild-type RNF146 3′UTR expression but did not affect respective mutant RNF146 3′UTR (*Figure 5E*), indicating that miR-306 and miR-79 directly target RNF146 3′UTR (*Figure 5D*). We also confirmed that overexpression of miR-306 or miR-79 reduced the endogenous levels of RNF146 protein (*Figure 5F*, quantified in *Figure 5G*) and that suppression of miR-306 and miR-79 functions by using miRNA sponges increased the endogenous levels of RNF146 protein in the adult eyes (*Figure 5—figure supplement 3A*, quantified in *Figure 5—figure supplement 3B*).

We next investigated whether RNF146 is the responsible target of miR-306 and miR-79 for the enhancement of JNK signaling. We found that, while knockdown of RNF146 did not affect normal tissue growth (*Figure 5—figure supplement 4A and B*, quantified in *Figure 5—figure supplement 4C*), it significantly suppressed Ras$^{V12}$/*dlg*$^{-/-}$ tumor growth (*Figure 5H–J*, quantified in *Figure 5K*) and promoted elimination of *dlg*$^{-/-}$ clones (*Figure 5—figure supplement 4D and E*, quantified in *Figure 5—figure supplement 4F*). Although overexpression of one copy of miR-306 or miR-79 in

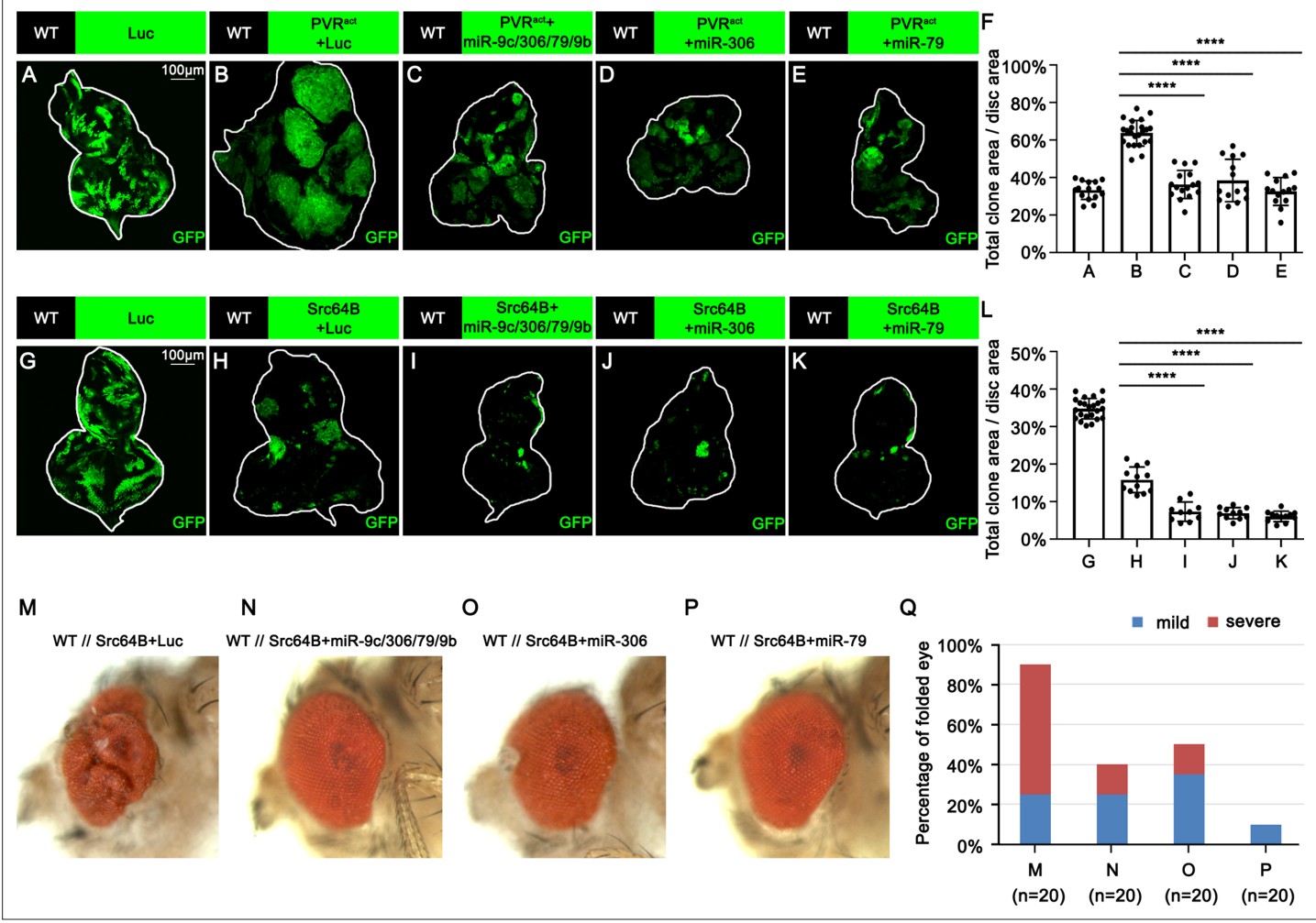

**Figure 4.** miR-306 and miR-79 suppress growth of multiple types of tumor models. (**A–E**) Eye-antennal disc bearing GFP-labeled clones of indicated genotypes (**A**, 5 days after egg laying, **B–E**, 7 days after egg laying). (**F**) Quantification of clone size (% of total clone area per disc area in eye-antennal disc) of (**A–E**). Error bars, SD; ****p<0.0001 by one-way ANOVA multiple-comparison test. (**G–K**) Eye-antennal disc bearing GFP-labeled clones of indicated genotypes (**G**, 5 days after egg laying, **H–K**, 6 days after egg laying). (**L**) Quantification of clone size (% of total clone area per disc area in eye-antennal disc) of (**G–K**). Error bars, SD; ****p<0.0001 by one-way ANOVA multiple-comparison test. (**M–P**) Adult eye phenotype of flies with indicated genotypes. (**Q**) Quantification of percentage of folded eye in (**M–P**). n = 20 for each group.

The online version of this article includes the following source data and figure supplement(s) for figure 4:

**Source data 1.** Quantitative data for *Figure 4*.

**Source data 2.** Genotypes for *Figure 4* and *Figure 4—figure supplements 1–3*.

**Figure supplement 1.** miR-306 and miR-79 promote multiple types of cell competition.

**Figure supplement 1—source data 1.** Quantitative data for *Figure 4—figure supplement 1*.

**Figure supplement 2.** miR-306 and miR-79 enhance JNK signaling in multiple types of tumors or cell competition models.

**Figure supplement 2—source data 1.** Quantitative data for *Figure 4—figure supplement 2*.

**Figure supplement 3.** miR-306 and miR-79 enhance normally occurring JNK activity.

**Figure supplement 3—source data 1.** Quantitative data for *Figure 4—figure supplement 3*.

the eyes had no significant effect on the number of dying cells or eye morphology (*Figure 3—figure supplement 2I–K*, quantified in *Figure 3—figure supplement 2L*, *Figure 5—figure supplement 4G–J*, quantified in *Figure 5—figure supplement 4N*), overexpression of 2×miR-306 or 2×miR-79 in the eyes significantly increased the number of dying cells (*Figure 5—figure supplement 4K–M*, quantified in *Figure 5—figure supplement 4N*) and resulted in a reduced-eye phenotype (*Figure 5—figure supplement 4O and Q*). Overexpression of RNF146 rescued the reduced-eye phenotype

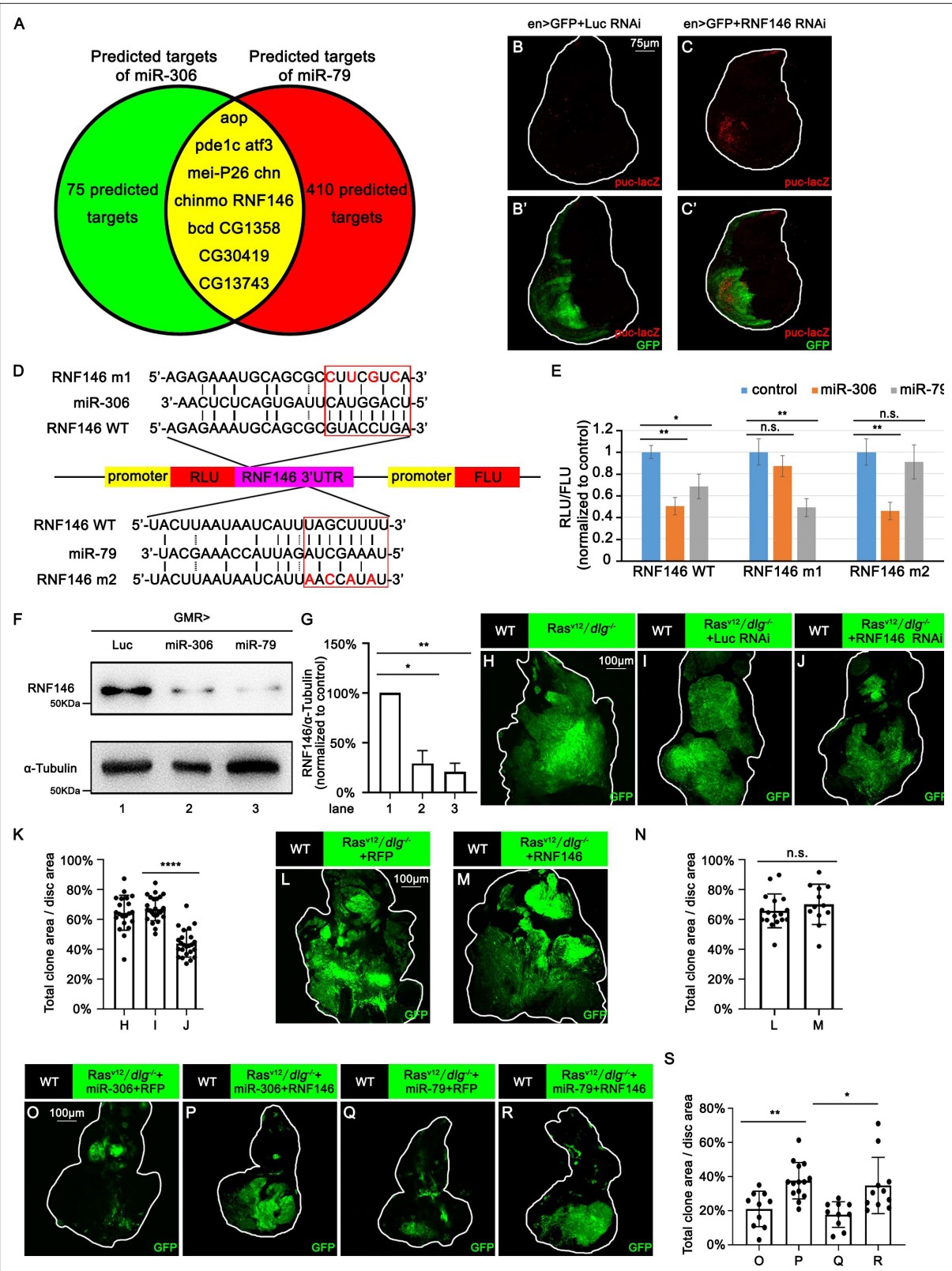

**Figure 5.** miR-306 and mir-79 suppress tumor growth and promote cell competition by targeting RNF146. Predicted targets of miR-306 and miR-79. (**B, C**) Wing disc of indicated genotypes with puc-lacZ background stained with anti-β-galactosidase antibody (**B,C** and **B', C'**, 5 or 6 days after egg laying). (**D**) Schematic of the wild-type and mutation-type 3'UTR vector with miRNA binding sites for miR-306 and miR-79, respectively. Red letters shows the mutation sites. Red box shows the seed sequence pairing region. (**E**) RLU/FLU rate from dual-luciferase assay. n = 3, error bars, SD; n.s., p>0.05 (not

*Figure 5 continued on next page*

*Figure 5 continued*

significant), **p<0.01 by two-tailed Student's *t*-test. (**F**) Lysates of adult heads of indicated genotypes were subjected to Western blots using indicated antibodies. (**G**) Quantification of relative levels of RNF146 protein in (**F**) from three independent experiments. Error bars, SD; *p<0.05, **p<0.01 by one-way ANOVA multiple-comparison test. (**H–J**) Eye-antennal disc bearing GFP-labeled clones of indicated genotypes (7 days after egg laying). (**K**) Quantification of clone size (% of total clone area per disc area in eye-antennal disc) of (**H–J**). Error bars, SD; ****p<0.0001 by two-tailed Student's *t*-test. (**L–M**) Eye-antennal disc bearing GFP-labeled clones of indicated genotypes (7 days after egg laying). (**N**) Quantification of clone size (% of total clone area per disc area in eye-antennal disc) of (**L–M**). Error bars, SD; n.s., p>0.05 (not significant) by two-tailed Student's *t*-test. (**O–R**) Eye-antennal disc bearing GFP-labeled clones of indicated genotypes (7 days after egg laying). (**S**) Quantification of clone size (% of total clone area per disc area in eye-antennal disc) of (**O–R**). Error bars, SD; *p<0.05, **p<0.01 by one-way ANOVA multiple-comparison test.

The online version of this article includes the following source data and figure supplement(s) for figure 5:

**Source data 1.** Quantitative data or raw data for *Figure 5*.

**Source data 2.** Genotypes for *Figure 5* and *Figure 5—figure supplements 1–6*.

**Figure supplement 1.** miR-306 and miR-79 promote JNK signaling in the wing disc.

**Figure supplement 2.** RNAis that target eight candidate genes do not induce JNK activation in the wing disc.

**Figure supplement 3.** Suppression of miR-306 and miR-79 functions promotes RNF146 protein level.

**Figure supplement 3—source data 1.** Quantitative data or raw data for *Figure 5—figure supplement 3*.

**Figure supplement 4.** miR-306 and miR-79 promote cell competition by targeting RNF146.

**Figure supplement 4—source data 1.** Quantitative data for *Figure 5—figure supplement 4*.

**Figure supplement 5.** Knocking down of RNF146 promotes JNK phosphorylation in Ras^V12/dlg^-/- tumor.

**Figure supplement 5—source data 1.** Quantitative data for *Figure 5—figure supplement 5*.

**Figure supplement 6.** miR-9 is predicted to target mammalian RNF146.

caused by overexpression of 2×miR-306 or 2×miR-79 in the eyes (*Figure 5—figure supplement 4O–R*, quantified in *Figure 5—figure supplement 4S*). Moreover, knocking down of RNF146 significantly enhanced Eiger-induced reduced-eye phenotype (*Figure 5—figure supplement 4T–W*, quantified in *Figure 5—figure supplement 4X*). Furthermore, although overexpression of RNF146 did not affect $Ras^{V12}/dlg^{-/-}$ tumor growth (*Figure 5L–M*, quantified in *Figure 5N*), overexpression of RNF146 weakened the tumor-suppressive effect of miR-306 or miR-79 on $Ras^{V12}/dlg^{-/-}$ tumors (*Figure 5O–R*, quantified in *Figure 5S*). The RNF146 overexpression also weakened the enhanced elimination of $dlg^{-/-}$ clones by miR-306 or miR-79 (*Figure 5—figure supplement 4Y–AB*, quantified in *Figure 5—figure supplement 4AC*). Similarly to ectopic expression of miR-306 or miR-79, knocking down of RNF146 enhanced JNK activity in $Ras^{V12}/dlg^{-/-}$ tumors (*Figure 5—figure supplement 5A and B*, quantified in *Figure 5—figure supplement 5C*). Together, these data indicate that miR-306 and miR-79 directly target RNF146 mRNA, thereby enhancing JNK signaling activity and thus exerting the tumor-suppressive effects.

## RNF146 promotes Tnks degradation

We next investigated the mechanism by which downregulation of RNF146 by miR-306 or miR-79 enhances JNK signaling activity. It has been shown in *Drosophila* embryos, larvae, wing discs, and adult eyes that loss of RNF146 upregulates the protein levels of Tnks (*Gultekin and Steller, 2019*; *Wang et al., 2019*), a poly-ADP-ribose polymerase that directly mediates poly-ADP ribosylation of JNK, which triggers K63-linked poly-ubiquitination of JNK and thereby promotes JNK-dependent apoptosis in *Drosophila* (*Feng et al., 2018*; *Li et al., 2018*). In addition, loss of RNF146 was shown to enhance rough-eye phenotype caused by Tnks overexpression (*Gultekin and Steller, 2019*). These observations raise the possibility that downregulation of RNF146 by miR-306 or miR-79 enhances JNK signaling via upregulation of Tnks. Indeed, as reported previously (*Feng et al., 2018*; *Li et al., 2018*), Western blot analysis revealed that overexpression of Tnks induces phosphorylation of JNK (JNK activation) in S2 cells (*Figure 6A*, lane 2 vs. lane 1, quantified in *Figure 6C*). Notably, coexpression of RNF146 significantly downregulated Tnks protein level and suppressed Tnks-induced JNK phosphorylation (*Figure 6A*, lane 3 vs. lane 2, quantified in *Figure 6B and C*). Moreover, knocking down of RNF146 or overexpression of miR-306 or miR-79 significantly upregulated Tnks protein level and promoted JNK phosphorylation (*Figure 6D*, quantified in *Figure 6E and F*, *Figure 6—figure supplement 1A*, quantified in *Figure 6—figure supplement 1B and C*). These data support the notion

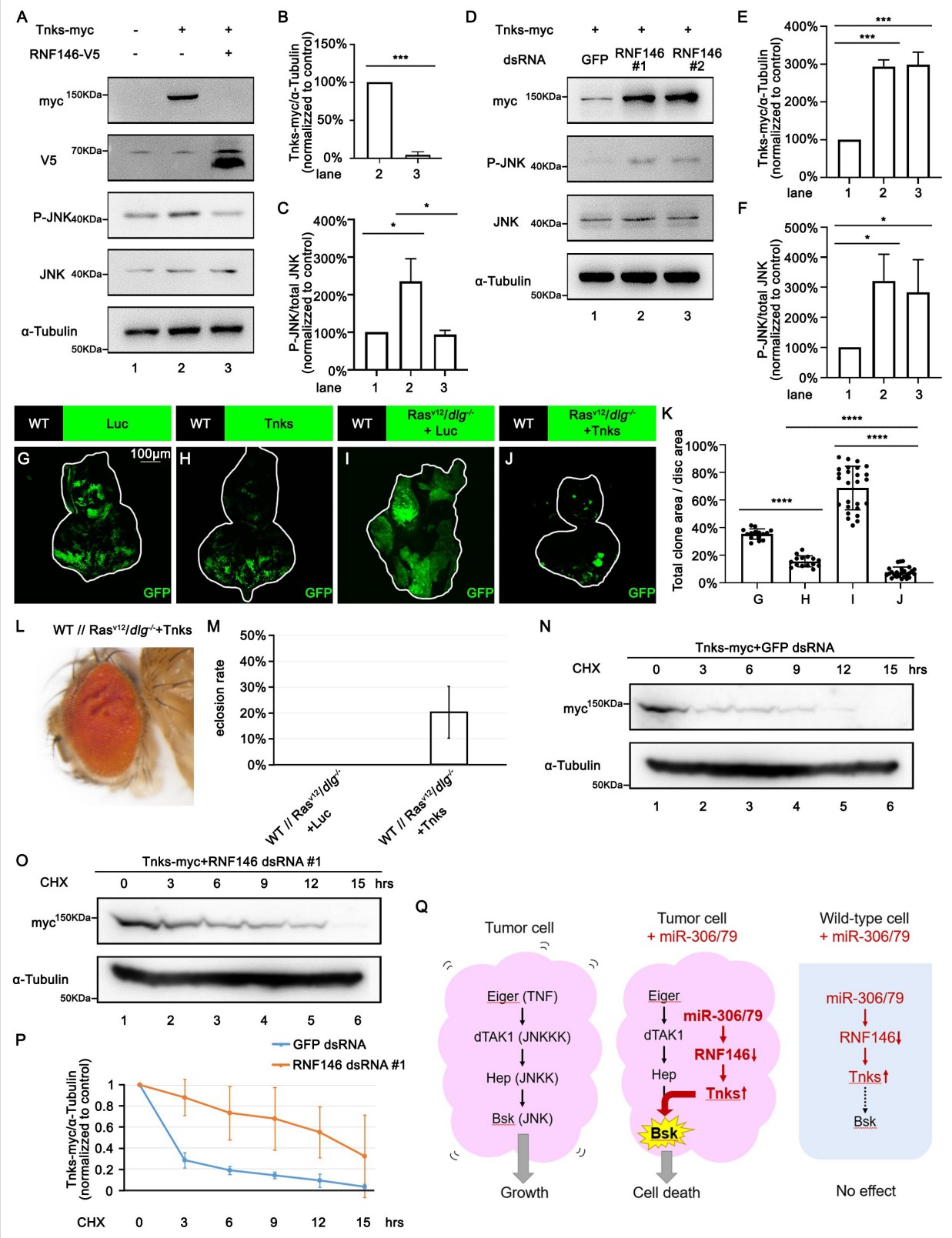

**Figure 6.** RNF146 promotes poly-ubiquitination and degradation of Tnks. (**A**) *Drosophila* S2 cells were transfected with plasmids expressing indicated proteins. Cell lysates were subjected to Western blots using indicated antibodies. (**B**) Quantification of relative Tnks-myc levels in (**A**) from three independent experiments. Error bars, SD; ***p<0.001 by two-tailed Student's *t*-test. (**C**) Quantification of relative p-JNK levels in (**A**) from three independent experiments. Error bars, SD; *p<0.05 by one-way ANOVA multiple-comparison test. (**D**) *Drosophila* S2 cells were transfected with plasmid

*Figure 6 continued on next page*

*Figure 6 continued*

expressing indicated protein and dsRNA targeting indicated gene. (**E, F**) Quantification of relative Tnks-myc levels (**E**) and p-JNK (**F**) levels in (**D**) from three independent experiments. Error bars, SD; *p<0.05, ***p<0.001 by one-way ANOVA multiple-comparison test. (**G–J**) Eye-antennal disc bearing GFP-labeled clones of indicated genotypes (**G, H**, 5 days after egg laying, **I, J**, 7 days after egg laying). (**K**) Quantification of clone size (% of total clone area per disc area in eye-antennal disc) of (**G–J**). Error bars, SD; ****p<0.0001 by one-way ANOVA multiple-comparison test. (**L**) Adult eye phenotype of flies with indicated genotypes. (**M**) Eclosion rate of flies with indicated genotypes. Data from three independent experiment, n > 30 for each group in one experiment; error bars, SD. (**N, O**) *Drosophila* S2 cells were transfected with plasmid expressing indicated protein and dsRNA targeting indicated gene. After 36 hr, cells were treated with 50 µg/ml cycloheximide (CHX) for the indicated periods. Cell lysates were subjected to Western blots using indicated antibodies. (**P**) Quantification of relative Tnks-myc levels in (**N, O**) from three independent experiments. Error bars, SD. (**Q**) A model for tumor elimination by miR-306/79. Tumor cell with elevated canonical JNK signaling via Eiger/TNF, dTAK1/JNKKK, and Hep/JNKK grows in a Bsk/JNK-dependent manner. Overexpression of miR-306 or miR-79 in JNK-activated tumor cell results in overactivation of JNK signaling to the lethal level via RNF146-Tnks-mediated noncanonical JNK-activating signaling. Overexpression of miR-306 or miR-79 in normal cells has no significant effect on JNK signaling.

The online version of this article includes the following source data and figure supplement(s) for figure 6:

**Source data 1.** Quantitative data or raw data for *Figure 6* (part 1).

**Source data 2.** Quantitative data or raw data for *Figure 6* (part 2).

**Source data 3.** Genotypes for *Figure 6*.

**Figure supplement 1.** miR-306 and miR-79 increase Tnks protein level.

**Figure supplement 1—source data 1.** Source data for *Figure 6—figure supplement 1*.

that downregulation of RNF146 or overexpression of miR-306 or miR-79 enhances JNK activation via upregulation of Tnks. Indeed, overexpression of Tnks was sufficient to suppress growth of either normal tissues or Ras^V12^/*dlg*^-/-^ tumors (*Figure 6G–J*, quantified in *Figure 6K*). Due to the fact that overexpression of Tnks alone resulted in larger clone size than Ras^V12^/*dlg*^-/-^+Tnks clone (*Figure 6H and J*, quantified in *Figure 6K*), our data support the notion that Tnks suppresses growth of Ras^V12^/*dlg*^-/-^ tumors by cooperating with JNK signaling. Consistent with the data shown above, overexpression of Tnks rescued the lethality of flies bearing Ras^V12^/*dlg*^-/-^ tumors in the eye-antennal discs (*Figure 6L and M*).

Finally, we sought to clarify the mechanism by which downregulation of RNF146 upregulates Tnks. A pervious study has shown that Tnks protein levels were significantly higher in *Rnf146* mutant background than in wild-type (*Gultekin and Steller, 2019*). However, this upregulation of Tnks can be caused by either elevated Tnks protein synthesis or reduced Tnks protein degradation. We thus examined the possibility that RNF146 promotes degradation of Tnks. Blocking new protein synthesis in S2 cells by the protein synthesis inhibitor cycloheximide (CHX) resulted in a time-dependent depletion of Tnks protein with a half-life of less than 3 hr (*Figure 6N*, quantified in *Figure 6P*). This depletion of Tnks was significantly retarded when RNF146 was knocked down (*Figure 6O*, quantified in *Figure 6P*). These data indicate that endogenous RNF146 promotes degradation of Tnks protein. Taken together, our data show that miR-306 or miR-79 directly targets RNF146, thereby leading to elevation of Tnks protein that induces noncanonical activation of JNK signaling (*Figure 6Q*).

## Discussion

In this study, we have identified the clustered miRNAs miR-306 and miR-79 as novel antitumor miRNAs that selectively eliminate JNK-activated tumors from *Drosophila* imaginal epithelia. Mechanistically, miR-306 and miR-79 directly target RNF146, an E3 ligase that promotes degradation of a poly-ADP-ribose polymerase Tnks, thereby leading to upregulation of Tnks and thus promoting JNK activation (*Figure 6K*). Importantly, this noncanonical mode of JNK activation has only a weak effect on normal tissue growth but it strongly blocks tumor growth by overactivating JNK signaling when tumors already possess elevated JNK signaling via the canonical JNK pathway (*Figure 6K*). Given that tumors or premalignant mutant cells often activate canonical JNK signaling, miR-306 and miR-79 can be novel ideal targets of cancer therapy.

Our study identified several putative co-target genes of miR-306 and miR-79 (*Figure 5A*). Interestingly, some of these genes (*Atf3*, *chinmo*, and *chn*) have been reported to be involved in tumor growth in *Drosophila*. *Atf3* encodes an AP-1 transcription factor that was shown to be a polarity-loss responsive gene acting downstream of the membrane-associated Scrib polarity complex (*Donohoe*

*et al., 2018*). Knockdown of *Atf3* suppresses growth and invasion of Ras[V12]/scrib[-/-] tumors in eye-antennal discs (*Atkins et al., 2016*). *Chinmo* is a BTB-zinc finger oncogene that is upregulated by JNK signaling in tumors (*Doggett et al., 2015*). Although loss of *chinmo* does not significantly suppress tumor growth, overexpression of *chinmo* with Ras[V12] or an activated Notch is sufficient to promote tumor growth in eye-antennal discs (*Doggett et al., 2015*). *Chn* encodes a zinc finger transcription factor that cooperates with *scrib[-/-]* to promote tumor growth (*Turkel et al., 2013*). Although we found that knockdown of these genes did not activate JNK signaling, it is possible that these putative target genes also contribute to the miR-306/miR-79-induced tumor suppression.

Intriguingly, it has been reported that miR-79 is downregulated in Ras[V12]/lgl-RNAi tumors in *Drosophila* wing discs (*Shu et al., 2017*). Given that miR-306 is located in the same miRNA cluster with miR-79, it is highly possible that miR-306 is also downregulated in tumors. This suggests that tumors have the mechanism that downregulates antitumor miRNAs for their survival and growth. Future studies on the mechanism of how tumors regulate these miRNAs would provide new understanding of tumor biology.

Our study uncovered the miR-306/79-RNF146-Tnks axis as noncanonical JNK enhancer that selectively eliminates JNK-activated tumors in *Drosophila*. Considering that miR-9, the mammalian homolog of miR-79, is predicted to target mammalian RNF146 (*Figure 5—figure supplement 6*) and that JNK signaling is highly conserved throughout evolution, it opens up the possibility of developing a new miRNA-based strategy against cancer.

## Materials and methods

### Fly stocks

All flies used were reared at 25°C on a standard cornmeal/yeast diet. Fluorescently labeled mitotic clones were produced in larval imaginal discs using the following strains: Tub-Gal80, FRT40A; eyFLP6, Act>y[+]>Gal4, UAS-GFP (40A tester), FRT42D, Tub-Gal80/CyO; eyFLP6, Act>y[+]>Gal4, UAS-GFP (42D tester), Tub-Gal80, FRT19A; eyFLP5, Act>y[+]>Gal4, UAS-GFP (19A tester #1), Tub-Gal80, FRT19A; eyFLP6, Act>y[+]>Gal4, UAS-GFP (19A tester #2). Additional strains used are the following: dlg[m52] (*Goode and Perrimon, 1997*), puc-lacZ (*Igaki et al., 2006*), UAS-Ras[v12] (*Igaki et al., 2006*), UAS-Bsk[DN] (*Adachi-Yamada et al., 1999*), UAS-Src64B (*Wills et al., 1999*), Hel25E[ccp-8] (*Nagata et al., 2019*), Mahj[1] (*Tamori et al., 2010*), UAS-N[act] (*Hori et al., 2004*), UAS-RNF146 (*Gultekin and Steller, 2019*); lgl[4] (BDSC #36289), UAS-p35 (BDSC #5073), UAS-PVR[act] (BDSC #58496), UAS-Yki[S168A] (BDSC #28836), UAS-Luciferase (BDSC #35788), UAS-RFP (BDSC #30556), UAS-bantam (BDSC #60672), UAS-miR-9c,306,79,9b (BDSC #41156), UAS-miR-79 (BDSC #41145), UAS-miR-2a-2,2a-1,2b-2 (BDSC #59849), UAS-miR-2b-1 (BDSC #41128), UAS-miR-7 (BDSC #41137), UAS-miR-8 (BDSC #41176), UAS-miR-9a (BDSC #41138), UAS-miR-9b (BDSC #41131), UAS-miR-9c (BDSC #41139), UAS-miR-11 (BDSC #59865), UAS-miR-12 (BDSC #41140), UAS-miR-13a,13b-1,2c (BDSC #64097), UAS-miR-13b-2 (BDSC #59867), UAS-miR-14 (BDSC #41178), UAS-miR-34 (BDSC #41158), UAS-miR-92a (BDSC #41153), UAS-miR-124 (BDSC #41126), UAS-miR-184 (BDSC #41174), UAS-miR-252 (BDSC #41127), UAS-miR-276a (BDSC #41143), UAS-miR-276b (BDSC #41162), UAS-miR-278 (BDSC #41180), UAS-miR-279 (BDSC #41147), UAS-miR-282 (BDSC #41165), UAS-miR-305 (BDSC #41152), UAS-miR-310 (BDSC #41155), UAS-miR-317 (BDSC #59913), UAS-miR-958 (BDSC #41222), UAS-miR-975,976,977 (BDSC #60635), UAS-miR-981 (BDSC #60639), UAS-miR-984 (BDSC #41224), UAS-miR-988 (BDSC #41196), UAS-miR-995 (BDSC #41199), UAS-miR-996 (BDSC #60653), UAS-miR-998 (BDSC #63043), UAS-miR-306-sponge (BDSC #61424), UAS-miR-79-sponge (BDSC #61387), UAS-Luciferase RNAi (BDSC #31603), UAS-aop RNAi (BDSC #34909), UAS-pde1c RNAi (BDSC #55925), UAS-atf3 RNAi (BDSC #26741), UAS-mei-P26 RNAi (BDSC #57268), UAS-chn RNAi (BDSC #26779), UAS-chinmo RNAi (BDSC #26777), UAS-RNF146 RNAi (BDSC #40882), UAS-bcd RNAi (BDSC #33886) and UAS-CG1358 RNAi (BDSC #64848) from Bloomington *Drosophila* Stock Center; UAS-miR-306 (FlyORF #F002214) from FlyORF; UAS-Tnks from Core Facility of *Drosophila* Resource and Technology, Center for Excellence in Molecular Cell Science, Chinese Academy of Sciences.

### Clone size measurement

Eye-antennal disc images were taken with a Leica SP8 confocal microscope or Olympus Fluoview FV3000 confocal microscope. To measure clone size, ImageJ (Fiji) software was used to determine the

threshold of the fluorescence. Total clone area/disc area (%) in the eye-antennal disc was calculated using ImageJ and Prism 8 (GraphPad).

## Histology

Larval tissues were stained with standard immunohistochemical procedures using rabbit anti-phospho-JNK polyclonal antibody (Cell Signaling Technology, Cat #4668, 1:100), chicken anti-β-galactosidase antibody (Abcam, Cat #ab9361, 1:1000), rabbit anti-Cleaved *Drosophila* Dcp-1 (Asp216) antibody (Cell Signaling Technology, Cat #9578, 1:100), goat anti-rabbit secondary antibody, Alexa Fluor 647 (Thermo Fisher Scientific, Cat #A32733, 1:250) or goat anti-chicken secondary antibody, Alexa Fluor 647 (Thermo Fisher Scientific, Cat #A21449, 1:250). Samples were mounted with DAPI-containing SlowFade Gold Antifade Reagent (Thermo Fisher Scientific, Cat #S36937). Images were taken with a Leica SP8 confocal microscope. The cleaved Dcp-1 positive cell number and the P-JNK-positive area was calculated using ImageJ and Prism 8 (GraphPad).

## Plasmid and in vitro transcription of dsRNA

pAc5.1/V5-His vector (Thermo Fisher Scientific, Cat #V411020) was used to construct plasmids for expressing proteins or miRNAs in *Drosophila* S2 cells. The RNF146 or Tnks ORF was amplified from fly cDNAs via PCR. The RNF146 ORF was cloned into the EcoRI-XhoI site of the pAc5.1/V5-His vector. The Tnks ORF carrying a myc tag at its 5′-end was cloned into the KpnI-XhoI site of the pAc5.1/V5-His vector. Extended region of miR-306 (–184 to +136) or miR-79 (–124 to +131) was amplified from fly cDNAs via PCR and cloned into the KpnI-EcoRI site of the pAc5.1/V5-His vector.

RNF146 dsRNA #1 and #2, respectively, targeting the 1–318 and 319–667 region of RNF146 ORF, used for RNF146 RNAi were transcribed in vitro using T7 RNA polymerase (Promega, Cat #P2075) at 37°C for 4 hr from the PCR products.

## Cell culture and transfection

*Drosophila* S2-ATCC cells (RRID:CVCL_Z232) was obtained from American Type Culture Collection (ATCC). Its identity was confirmed by visual inspection of the cell morphology and its growth kinetics in Schneider's *Drosophila* medium (Thermo Fisher Scientific, Cat #21720024)/10% fetal bovine serum (FBS) and penicillin/streptomycin. A mycoplasma test is usually not done for S2 cells.

For transfection assay, S2 cells were plated in 100 mm plates or six-well plates and grown overnight to reach 70% confluence. After that, DNA plasmids or dsRNAs were transfected into the cells using FuGene HD transfection reagent (Promega, Cat #PRE2311) according to the manufacturer's protocol. The protein synthesis inhibitor CHX (Santa Cruz Biotechnology, Cat #SC-3508) was used at 50 µg/ml.

## Western blots

Cultured *Drosophila* S2 cells were harvested and then lysed in cell lysis buffer. The cell lysates were then subjected to SDS-PAGE, followed by Western blots using anti-α-tubulin monoclonal antibody (Sigma-Aldrich, Cat #T5168, 1:5000), anti-phospho-JNK polyclonal antibody (Cell Signaling Technology, Cat #9251, 1:1000), anti-JNK monoclonal antibody (Santa Cruz Biotechnology, Cat #sc-7345, 1:1000), anti-RNF146 polyclonal antibody (raised in rabbits against the peptide HSGGGSGEDPAVGSC, GenScript antibody service, Nanjing, China, 1:2000), anti-V5 tag monoclonal antibody (Thermo Fisher Scientific, Cat #R960-25, 1:5000), anti-myc tag polyclonal antibody (MBL, Code #562, 1:1000), anti-mouse IgG, HRP-linked antibody (Cell Signaling Technology, Cat #7076, 1:5000), or anti-rabbit IgG, HRP-linked antibody (Cell Signaling Technology, Cat #7074, 1:5000).

## Dual-luciferase reporter assay

The psiCHECK-2 vector (Promega, Cat #C8021) was used to construct plasmids for dual-luciferase reporter assay. RNF146 3′UTR or its mutant was cloned into the XhoI-NotI site of the psiCHECK-2 vector. Renilla luciferase activity and firefly luciferase activity were measured using GloMax-Multi Jr Single-Tube Multimode Reader (Promega) according to the manufacturer's protocol.

## Statistical analysis

When comparing two groups, statistical significance was tested using a Student's *t*-test. When comparing multiple groups, statistical significance was tested using a one-way ANOVA multiple-comparison test.

In all figures, significance is indicated as follows: n.s. (not significant), $p > 0.05$, *$p < 0.05$, **$p < 0.01$, ***$p < 0.001$, and ****$p < 0.0001$.

## Acknowledgements

We thank M Matsuoka and K Gomi for technical support, Hermann Steller, the Bloomington *Drosophila* Stock Center, the National Institute of Genetics Stock Center (NIG-FLY), the *Drosophila* Genomics and Genetic Resources (DGGR, Kyoto Institute of Technology), the Vienna *Drosophila* Resource Center (VDRC), and the Core Facility of *Drosophila* Resource and Technology, Center for Excellence in Molecular Cell Science, Chinese Academy of Sciences for fly stocks and reagents. We also thank the members of the Igaki laboratory for discussions. This work was supported by grants from the MEXT/JSPS KAKENHI (grant numbers 20H05320, 21H05284, and 21H05039) to TI, Japan Agency for Medical Research and Development (Project for Elucidating and Controlling Mechanisms of Aging and Longevity; grant number 20gm5010001) to TI, the Takeda Science Foundation to TI, the Fundamental Research Funds for the Central Universities, Sun Yat-sen University to ZW (grant number 22hytd05), and the Naito Foundation to TI. ZW was supported by JSPS Postdoctoral Fellowships for Research in Japan, and XX was supported by China Scholarship Council for Research in Japan.

## Additional information

### Funding

| Funder | Grant reference number | Author |
|---|---|---|
| MEXT/JSPS KAKENHI | 20H05320 | Tatsushi Igaki |
| MEXT/JSPS KAKENHI | 21H05284 | Tatsushi Igaki |
| MEXT/JSPS KAKENHI | 21H05039 | Tatsushi Igaki |
| Japan Agency for Medical Research and Development | Project for Elucidating and Controlling Mechanisms of Aging and Longevity | Tatsushi Igaki |
| Japan Agency for Medical Research and Development | 20gm5010001 | Tatsushi Igaki |
| Takeda Science Foundation | | Tatsushi Igaki |
| Fundamental Research Funds for the Central Universities | Sun Yat-sen University | Zhaowei Wang |
| Fundamental Research Funds for the Central Universities | 22hytd05 | Zhaowei Wang |
| Naito Foundation | | Tatsushi Igaki |
| Japan Society for the Promotion of Science | Postdoctoral Fellowships for Research in Japan | Zhaowei Wang |
| China Scholarship Council | for Research in Japan | Xiaoling Xia |

The funders had no role in study design, data collection and interpretation, or the decision to submit the work for publication.

### Author contributions

Zhaowei Wang, Conceptualization, Formal analysis, Investigation, Visualization, Methodology, Writing - original draft, Writing - review and editing; Xiaoling Xia, Conceptualization, Formal analysis, Investigation, Visualization, Writing - original draft; Jiaqi Li, Investigation, Methodology; Tatsushi Igaki, Conceptualization, Formal analysis, Supervision, Funding acquisition, Investigation, Visualization, Writing - original draft, Project administration, Writing - review and editing

**Author ORCIDs**
Zhaowei Wang (iD) http://orcid.org/0000-0003-4491-8502
Tatsushi Igaki (iD) http://orcid.org/0000-0001-5839-9526

**Decision letter and Author response**
Decision letter https://doi.org/10.7554/eLife.77340.sa1
Author response https://doi.org/10.7554/eLife.77340.sa2

## Additional files

### Supplementary files
• Transparent reporting form

### Data availability
All relevant data are within the paper and its Supporting Information files. All the numerical data that are represented as a graph in a figure are provided in the Source Data file.

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

# Appendix 1

## Appendix 1—key resources table

| Reagent type (species) or resource | Designation | Source or reference | Identifiers | Additional information |
|---|---|---|---|---|
| Genetic reagent (*Drosophila melanogaster*) | dlg[m52] | PMID:9334318 | N/A | |
| Genetic reagent (*D. melanogaster*) | puc-lacZ | PMID:16753569 | N/A | |
| Genetic reagent (*D. melanogaster*) | UAS-Ras[v12] | PMID:16753569 | N/A | |
| Genetic reagent (*D. melanogaster*) | UAS-Bsk[DN] | PMID:10490662 | N/A | |
| Genetic reagent (*D. melanogaster*) | UAS-Src64B | PMID:10069336 | N/A | |
| Genetic reagent (*D. melanogaster*) | Hel25E[ccp-8] | PMID:31543447 | N/A | |
| Genetic reagent (*D. melanogaster*) | Mahj[1] | PMID:20644714 | N/A | |
| Genetic reagent (*D. melanogaster*) | UAS-N[act] | PMID:15496440 | N/A | |
| Genetic reagent (*D. melanogaster*) | UAS-RNF146 | PMID:30796047 | N/A | |
| Genetic reagent (*D. melanogaster*) | lgl[4] | Bloomington Drosophila Stock Center | BDSC:36289 | |
| Genetic reagent (*D. melanogaster*) | UAS-p35 | Bloomington Drosophila Stock Center | BDSC:5073 | |
| Genetic reagent (*D. melanogaster*) | UAS-PVR[act] | Bloomington Drosophila Stock Center | BDSC:58496 | |
| Genetic reagent (*D. melanogaster*) | UAS-Yki[S168A] | Bloomington Drosophila Stock Center | BDSC:28836 | |
| Genetic reagent (*D. melanogaster*) | UAS-Luciferase | Bloomington Drosophila Stock Center | BDSC:35788 | |
| Genetic reagent (*D. melanogaster*) | UAS-RFP | Bloomington Drosophila Stock Center | BDSC:30556 | |
| Genetic reagent (*D. melanogaster*) | UAS-bantam | Bloomington Drosophila Stock Center | BDSC:60672 | |
| Genetic reagent (*D. melanogaster*) | UAS-miR-9c,306,79,9b | Bloomington Drosophila Stock Center | BDSC:41156 | |
| Genetic reagent (*D. melanogaster*) | UAS-miR-79 | Bloomington Drosophila Stock Center | BDSC:41145 | |
| Genetic reagent (*D. melanogaster*) | UAS-miR-2a-2,2a-1,2b-2 | Bloomington Drosophila Stock Center | BDSC:59849 | |
| Genetic reagent (*D. melanogaster*) | UAS-miR-2b-1 | Bloomington Drosophila Stock Center | BDSC:41128 | |
| Genetic reagent (*D. melanogaster*) | UAS-miR-7 | Bloomington Drosophila Stock Center | BDSC:41137 | |
| Genetic reagent (*D. melanogaster*) | UAS-miR-8 | Bloomington Drosophila Stock Center | BDSC:41176 | |
| Genetic reagent (*D. melanogaster*) | UAS-miR-9a | Bloomington Drosophila Stock Center | BDSC:41138 | |
| Genetic reagent (*D. melanogaster*) | UAS-miR-9b | Bloomington Drosophila Stock Center | BDSC:41131 | |

*Appendix 1 Continued on next page*

*Appendix 1 Continued*

| Reagent type (species) or resource | Designation | Source or reference | Identifiers | Additional information |
|---|---|---|---|---|
| Genetic reagent (*D. melanogaster*) | UAS-miR-9c | Bloomington Drosophila Stock Center | BDSC:41139 | |
| Genetic reagent (*D. melanogaster*) | UAS-miR-11 | Bloomington Drosophila Stock Center | BDSC:59865 | |
| Genetic reagent (*D. melanogaster*) | UAS-miR-12 | Bloomington Drosophila Stock Center | BDSC:41140 | |
| Genetic reagent (*D. melanogaster*) | UAS-miR-13a,13b-1,2c | Bloomington Drosophila Stock Center | BDSC:64097 | |
| Genetic reagent (*D. melanogaster*) | UAS-miR-13b-2 | Bloomington Drosophila Stock Center | BDSC:59867 | |
| Genetic reagent (*D. melanogaster*) | UAS-miR-14 | Bloomington Drosophila Stock Center | BDSC:41178 | |
| Genetic reagent (*D. melanogaster*) | UAS-miR-34 | Bloomington Drosophila Stock Center | BDSC:41158 | |
| Genetic reagent (*D. melanogaster*) | UAS-miR-92a | Bloomington Drosophila Stock Center | BDSC:41153 | |
| Genetic reagent (*D. melanogaster*) | UAS-miR-124 | Bloomington Drosophila Stock Center | BDSC:41126 | |
| Genetic reagent (*D. melanogaster*) | UAS-miR-184 | Bloomington Drosophila Stock Center | BDSC:41174 | |
| Genetic reagent (*D. melanogaster*) | UAS-miR-252 | Bloomington Drosophila Stock Center | BDSC:41127 | |
| Genetic reagent (*D. melanogaster*) | UAS-miR-276a | Bloomington Drosophila Stock Center | BDSC:41143 | |
| Genetic reagent (*D. melanogaster*) | UAS-miR-276b | Bloomington Drosophila Stock Center | BDSC:41162 | |
| Genetic reagent (*D. melanogaster*) | UAS-miR-278 | Bloomington Drosophila Stock Center | BDSC:41180 | |
| Genetic reagent (*D. melanogaster*) | UAS-miR-279 | Bloomington Drosophila Stock Center | BDSC:41147 | |
| Genetic reagent (*D. melanogaster*) | UAS-miR-282 | Bloomington Drosophila Stock Center | BDSC:41165 | |
| Genetic reagent (*D. melanogaster*) | UAS-miR-305 | Bloomington Drosophila Stock Center | BDSC:41152 | |
| Genetic reagent (*D. melanogaster*) | UAS-miR-310 | Bloomington Drosophila Stock Center | BDSC:41155 | |
| Genetic reagent (*D. melanogaster*) | UAS-miR-317 | Bloomington Drosophila Stock Center | BDSC:59913 | |
| Genetic reagent (*D. melanogaster*) | UAS-miR-958 | Bloomington Drosophila Stock Center | BDSC:41222 | |
| Genetic reagent (*D. melanogaster*) | UAS-miR-975,976,977 | Bloomington Drosophila Stock Center | BDSC:60635 | |
| Genetic reagent (*D. melanogaster*) | UAS-miR-981 | Bloomington Drosophila Stock Center | BDSC:60639 | |
| Genetic reagent (*D. melanogaster*) | UAS-miR-984 | Bloomington Drosophila Stock Center | BDSC:41224 | |
| Genetic reagent (*D. melanogaster*) | UAS-miR-988 | Bloomington Drosophila Stock Center | BDSC:41196 | |
| Genetic reagent (*D. melanogaster*) | UAS-miR-995 | Bloomington Drosophila Stock Center | BDSC:41199 | |

*Appendix 1 Continued on next page*

*Appendix 1 Continued*

| Reagent type (species) or resource | Designation | Source or reference | Identifiers | Additional information |
|---|---|---|---|---|
| Genetic reagent (*D. melanogaster*) | UAS-miR-996 | Bloomington Drosophila Stock Center | BDSC:60653 | |
| Genetic reagent (*D. melanogaster*) | UAS-miR-998 | Bloomington Drosophila Stock Center | BDSC:63043 | |
| Genetic reagent (*D. melanogaster*) | UAS-miR-306-sponge | Bloomington Drosophila Stock Center | BDSC:61424 | |
| Genetic reagent (*D. melanogaster*) | UAS-miR-79-sponge | Bloomington Drosophila Stock Center | BDSC:61387 | |
| Genetic reagent (*D. melanogaster*) | UAS-Luciferase RNAi | Bloomington Drosophila Stock Center | BDSC:31603 | |
| Genetic reagent (*D. melanogaster*) | UAS-aop RNAi | Bloomington Drosophila Stock Center | BDSC:34909 | |
| Genetic reagent (*D. melanogaster*) | UAS-pde1c RNAi | Bloomington Drosophila Stock Center | BDSC:55925 | |
| Genetic reagent (*D. melanogaster*) | UAS-atf3 RNAi | Bloomington Drosophila Stock Center | BDSC:26741 | |
| Genetic reagent (*D. melanogaster*) | UAS-mei-P26 RNAi | Bloomington Drosophila Stock Center | BDSC:57268 | |
| Genetic reagent (*D. melanogaster*) | UAS-chn RNAi | Bloomington Drosophila Stock Center | BDSC:26779 | |
| Genetic reagent (*D. melanogaster*) | UAS-chinmo RNAi | Bloomington Drosophila Stock Center | BDSC:26777 | |
| Genetic reagent (*D. melanogaster*) | UAS-RNF146 RNAi | Bloomington Drosophila Stock Center | BDSC:40882 | |
| Genetic reagent (*D. melanogaster*) | UAS-bcd RNAi | Bloomington Drosophila Stock Center | BDSC:33886 | |
| Genetic reagent (*D. melanogaster*) | UAS-CG1358 RNAi | Bloomington Drosophila Stock Center | BDSC:64848 | |
| Genetic reagent (*D. melanogaster*) | UAS-miR-306 | FlyORF | FlyORF: F002214 | |
| Genetic reagent (*D. melanogaster*) | UAS-Tnks | Core Facility of *Drosophila* Resource and Technology, Center for Excellence in Molecular Cell Science, Chinese Academy of Sciences | N/A | |
| Cell line (*D. melanogaster*) | S2 | ATCC | Cat #CRL-1963 | |
| Antibody | Anti-phospho-JNK (rabbit monoclonal) | Cell Signaling Technology | Cat #4668 | 1:100 |
| Antibody | Anti-β-galactosidase (chicken polyclonal) | Abcam | Cat #ab9361 | 1:1000 |
| Antibody | Anti-cleaved *Drosophila* Dcp-1 (Asp216) (rabbit polyclonal) | Cell Signaling Technology | Cat #9578 | 1:100 |
| Antibody | Goat anti-rabbit secondary antibody, Alexa Fluor 647 | Thermo Fisher Scientific | Cat #A32733 | 1:250 |
| Antibody | Goat anti-chicken secondary antibody, Alexa Fluor 647 | Thermo Fisher Scientific | Cat #A21449 | 1:250 |

*Appendix 1 Continued on next page*

*Appendix 1 Continued*

| Reagent type (species) or resource | Designation | Source or reference | Identifiers | Additional information |
|---|---|---|---|---|
| Antibody | Anti-α-tubulin (mouse monoclonal) | Sigma-Aldrich | Cat #T5168 | 1:5000 |
| Antibody | Anti-phospho-JNK (rabbit polyclonal) | Cell Signaling Technology | Cat #9251 | 1:1000 |
| Antibody | Anti-JNK (mouse monoclonal) | Santa Cruz Biotechnology | Cat #sc-7345 | 1:1000 |
| Antibody | Anti-RNF146 (rabbit polyclonal) | GenScript antibody service | N/A | Raised in rabbits against peptide HSGGGSGEDPAVGSC,1:2000 |
| Antibody | Anti-V5 tag (mouse monoclonal) | Thermo Fisher Scientific | Cat #R960-25 | 1:5000 |
| Antibody | Anti-myc tag (rabbit polyclonal) | MBL | Cat #562 | 1:1000 |
| Antibody | Horse anti-mouse IgG, HRP-linked antibody | Cell Signaling Technology | Cat #7076 | 1:5000 |
| Antibody | Goat anti-rabbit IgG, HRP-linked antibody | Cell Signaling Technology | Cat #7074 | 1:5000 |
| Commercial assay or kit | DAPI-containing SlowFade Gold Antifade Reagent | Thermo Fisher Scientific | Cat #S36937 | |
| Commercial assay or kit | FuGene HD transfection reagent | Promega | Cat #PRE2311 | |
| Other | CHX | Santa Cruz Biotechnology | Cat #SC-3508 | 50 µg/ml |

