## [Editor Report]

This article is valuable as it uncovers a previously unknown tumor-suppressor mechanism that eliminates *JNK*-activated *Drosophila* tumors. This mechanism is triggered by the overexpression of microRNAs that downregulate an E3 ubiquitin ligase RNF146, whose loss causes an increase in Tnks (poly-ADP-ribose polymerases) and *JNK* signaling. This tumor-suppressor mechanism has potential implications for the treatment of *JNK*-activated tumors. This article is of interest to people in the tumor suppressor, *JNK*, and miRNA fields, and the key claims are convincing and well supported by the data, and the authors use thoughtful and rigorous approaches.

---

## [Decision Letter]

**Decision letter after peer review:**

Thank you for submitting your article "Tumor elimination by clustered microRNAs miR-306 and miR-79 via non-canonical activation of *JNK* signaling" for consideration by *eLife*. Your article has been reviewed by 3 peer reviewers, including Erika A Bach as Reviewing Editor and (Reviewer #1), and the evaluation has been overseen by Utpal Banerjee as the Senior Editor.

Essential revisions:

1. Show whether 1x or 2x copies of miR-306 or miR-79 cause death in larval or pupal eye discs.

2. Assess whether Rnf146 protein increases in miR 9c/306/79/9b loss-of-function clones in the eye disc.

3. Ensure that the same number of UAS transgenes are present in genotypes that are being compared; this might result in repeating experiments.

4. Improve Western blots of pJNK by normalizing to the level of total *JNK* (not α-tubulin) in the sample and increasing sample size to at least three independent replicates. The Western blot data should be displayed as a graph of normalized p-*JNK* levels with error bars and statistics.

5. Improve the Rnf146 Western blot in Figure 5F. At a bare minimum, there should be at least three replicates and quantification in a graph.

6. Assess whether these miRNAs enhance normally-occurring *JNK* activity, for example during dorsal closure in the embryo.

7. Many controls are not shown. These include but are not limited to:

– Figure 1 needs wild-type control (Luc) clones and quantification.

– Figure 2 needs wild-type control (Luc) clones and quantification (to the block of panels in Figure 2A-E) and p35-expressing clones and quantification (to the block of panels in Figure 2. F-I).

– Figure 3 needs wild-type control clones (Luc alone), and p35-expressing clones the first block of panels (A-I). In the Q-X section, the authors should add Luc alone clones, bsk-DN clones, Ras/dlg clones with quantification. M-P should show control eyes (GMR/+).

– Figure 4 A-E and Figure 4-Figure Sup 1, wild-type control clones are required.

– Figure 5, Ras-Dlg and dRNF146-OE clones should be included.

– Figure 6C, D, should include Tnks alone expressing clones.

8. Show expression of *JNK* targets – TRE-reporters, pJNK, or puc-lacZ – in miR-306 and 79 clones and in PVR-act, Src64B, Hel25E, and Mahj clones.

9. Show that miR-306 and 79 enhance *JNK* activity in Ras-Dlg tumors by showing the expression of *JNK* targets as the TRE-reporters, pJNK, or puc-lacZ.

10. Show that Ras-Dlg+miR-306/79+dRNF146 clones are the same size as the Ras-Dlg clones. Currently, this does not seem to be the case (compare Fig5G with Figure 5K or M).

11. Show that *JNK* activity levels are upregulated when dRNF146 is downregulated in those tumors.

12. Show that Tnks overexpression alone does not affect normal growth in otherwise wild-type clones.

13. Show that miR-309/79 modulation affects Tnks protein levels.

*Reviewer #1 (Recommendations for the authors):*

In the current study, Wang and colleagues perform a microRNA (miR) screen to find miRs that can suppress tumors caused by RasV12 and loss of polarity genes (i.e., discs large (dlg) or lethal giant larva (lgl)). This screen identified 12 miRs that when overexpressed suppressed the growth of RasV12 dlg-/- (or lgl-/-) tumors in the eye disc. They went on to characterize a cluster called miR 9c/306/79/9b. Clonal overexpression of miR-306 or miR-79 suppressed the tumors but did not greatly suppress growth of wild-type tissue. The authors went on to show that clonal overexpression of miR-306 or of miR-79 caused cell death in the tumors but did not cause death in WT tissue. Further experiments revealed that these miRs interact with cell polarity and not with RasV12. The authors then show that these miRs cause up regulation of *JNK* signaling as inhibiting *JNK* signaling abrogated the tumor-suppressing ability of these miRs. These miRs also enhanced *JNK* signaling in several other kinds of tumors that depend on *JNK*. They use an algorithm targetscanfly to search for miR binding sites and found 11 mRNAs that were predicted to be targets of both miR-306 and miR-79. They were able to test nine of these genes through RNA interference and only one of them Rnf146 caused *JNK* activation when depleted. They used luciferase assays in cultured cells to show that there were two binding sites in the 3'UTR of Rnf146 for these miRs. Knockdown of Rnf146 did not block the growth of normal tissue but it did significantly suppress the growth of RasV12 dlg-/- mutant cells. Rnf146 is a ubiquitin ligase that interacts with tankyrases (TNKs) – poly-ADP-ribose polymerases – to target proteins for degradation, and the authors next examined the role of TNKs in their tumor model. They used biochemistry to show that TNKs activate *JNK* by phosphorylation in S2 cells and that RNF146 downregulates TNK protein as well as *JNK* activation. Overexpression of TNKs in tumors suppresses their growth. Finally, they used biochemistry to show that when they blocked protein synthesis and depleted Rnf146, TNK protein levels remained relatively stable. These data support their model that miR-306 and -79 directly target Rnf146, which results in elevated TNK and this induces *JNK* signaling to cause cell death.

The novelty of this study is finding the connection between miR-306 and miR-79 and Rnf146. By contrast, it was already known that TNKs activates *JNK* signaling (Feng et al. 2018; Li et al. 2018) and that Rnf146 degrades TNKs (Gultekin, Y., Steller, H., 2019, Figure S2E-F, PMID: 30796047). The lattermost result is not acknowledged by the authors. Gultekin and Steller mis-expressed TNKs in eye discs in a WT or an RNF146 mutant background (Iduna-/- which is adult viable) and then performed Western blots on eye discs. They found that TNK protein levels were significantly higher in the Rnf146 mutant background than in WT.

There are several issues that I think should be addressed.

Issues with experiments/text:

1. The authors should acknowledge the results of Gultekin and Steller in the manuscript and compare results.

2. The authors state that over-expression of miR-306 or miR-79 (presumably using 1 copy of the transgene) does not reduce growth or WT cells. However, the results in Figure 5, figure supplement 3G-J does not support this model. In these experiments, the authors over-expressed 2 copies of UAS-miR and adult eyes from GMR>2x miR-306 or GMR>2x miR-79 are noticeably smaller than WT and are rough. These results suggest that miR-306 and miR-79 do indeed affect WT cells, which then may make it less likely that miR over-expression can be used as cancer therapy. Can the authors examine death cells in GMR>2x miR-306 or GMR>2x miR-79 larval or pupal discs? Alternatively, what happens when only 1 copy of the UAS-miR is over-expressed?

3. The authors depleted Rnf146 from GMR>Eiger and saw an enhancement of the eye phenotype. I might be confused, but I thought that it would be better to over-express Rnf146 in GMR>Eiger and look for suppression of the small eye phenotype.

4. Most of the experiments are over-expression. Does miR-306 or miR-79 normally regulate levels of Rnf-146 protein? Can you make a clone of the miR 9c/306/79/9b and see Rnf146 protein levels increase?

5. The Rnf146 null allele is adult viable. It might be beyond the scope of this work, but if you made RasV12, dlg-/- clones in an Rnf146 null mutant, the tumors should significantly smaller than in a heterozygous background.

Issues with figures:

1. Figure 1, figure supplement 1, panel AI: the authors need to mention that the dashed horizontal line at 60% comes from the RasV12 dlg-/- clones in Figure 1F.

2. Figure 2: Please show Dcp-1 in red or magenta in panels A'-D' as it is very difficult to see white on green.

3. Figure 2: panel I – there should be the same number of UAS transgenes in all the genotypes.

4. 4. Figure 2, figure supplement 1: Please show Dcp-1 in red or magenta in panels A'-D' as it is very difficult to see white on green.

5. Figure 3: panels T and X, there should be the same number of UAS transgenes in all the genotypes.

6. Figure 3, panel J: The authors should show a close up of pJNK in WT wing discs with Luc clones as panel J".

7. Figure 3, figure supplement 2, panel A: The authors should show a close up of pJNK in WT wing discs with Luc clones as panel A".

8. Figure 3, figure supplement 2, panel D: There are several issues with this figure. (1) pJNK levels should be normalize to the level of total *JNK* in the sample. It is possible that the lower level of pJNK is a result of less *JNK* protein in GMR>Luc compared to *JNK* in GMR>miR-306 or GMR-miR-79 and knowing the total level of *JNK* protein in these cells would allow them to disprove this possibility. Currently, the authors normalize with α-tubulin. (2) The Western blot data should be displayed as a graph of normalized p-*JNK* levels with error bars and statistics.

9. Figure 5, panels B' and C': Please show puc-lacZ in red or magenta in panels A'-D' as it is very difficult to see white on green.

10. Figure 5, panel F: The Western blot data should be displayed as a graph of normalized dRNF146 levels with error bars and statistics.

11. Figure 5, figure supplement 1, panels A'-F': Please show puc-lacZ in red or magenta as it is very difficult to see white on green.

12. The source files are in a.gel format, which I cannot access. Would you please upload them as.tif or.jpeg?

13. Include a file with complete genotypes for all figures.

*Reviewer #2 (Recommendations for the authors):*

This is a very complete manuscript. It requires only very few revisions and is appropriate for eLife.

1. The authors showed that overexpression of miR-79 and -306 alone was not sufficient to induce a significant phenotype on eye morphology. However, JNK activity is usually not active in eye discs and therefore it cannot be further enhanced. Can expression of these miRNAs enhance normally occurring JNK activity during normal development? One example where the author can address this question is dorsal closure during embryogenesis.

2. The authors mention that Tankyrase promotes K63-polyubiquitylation of JNK. Is it known how Tankyrase mediates this effect? Even if not, it is worth mentioning this in the text.

3. The authors did an excellent job in quantifying every experiment shown. However, in the Method section, they did not explain how the expression levels (Dcp-1, pJNK, puc-lacZ, etc.) were measured and quantified.

*Reviewer #3 (Recommendations for the authors):*

The experiments presented lack some controls. This will allow a more complete comparison between the genetic conditions analyzed.

– Figure 1 should show, as panel A, wild type control clones (Luc alone). The quantification of those clones should be shown in panel F.

– Figure 2: wild type control clones should be added in the 1st block of results (A-E). p35-expressing clones should be used as controls in the second group of results (F-I).

– Figure 3: the first block of results (A-I) should also include wild type control clones (Luc alone), and p35-expressing clones. In the Q-X section, Luc alone clones, bsk-DN clones, Ras/dlg clones should be shown and quantified. M-P should show control eyes (GMR/+).

– Figure 4 A-E, and Figure 4-Figure Sup 1, wild type control clones are required.

– Figure 5, Ras-Dlg and dRNF146-OE clones should be included.

– Figure 6C, D, hould include Tnks alone expressing clones.

To conclude that "miR-306 and miR-79 suppress tumor growth by enhancing *JNK* signaling", the authors should show that miR-306 and 79 enhance *JNK* activity in tumors by showing the expression of *JNK* targets as the TRE-reporters, pJNK, or puc-lacZ.

The title of section "miR-306 and miR-79 enhance *JNK* signaling in different types of tumors" is misleading and confusing. Among the 4 conditions analyzed – namely, PVR-act, Src64B, Hel25E, and Mahj, only the PVR-act condition (although it lacks the proper control) seems to cause oncogenic growth. As previously mentioned, to claim that those miRNAs enhance *JNK* activity, the authors should show the expression of *JNK* targets as the TRE-reporters, pJNK, or puc-lacZ.

Figure 4 K-O. The authors show that "non-autonomous overgrowth of surrounding wild-type tissue by Src64B-overexpressing clones". Is this observation relevant for this manuscript? It seems to come out of the blue and I can´t find the connection with the rest of results presented here.

Related to the regulation of dRNF146 by miRNA-305/79, the results shown in D and E show that the predicted binding site is sensitive to the levels of those miRNAs. However, results showing that the proposed regulation is indeed taking place in vivo and is physiologically relevant are missing. The western presented in Figure 5F is very poor. Results with better quality need to be provided to show that convincingly. In line with that, showing the protein levels in miR-306 and 79-expressing clones in the eye disc will provide a convincing piece of data.

The authors claim: "Furthermore, overexpression of dRNF146 cancelled the tumor-suppressive effect of miR-306 or miR-79 on RasV12/dlg-/- tumors (Figure 5J-M, quantified in Figure 5N)." To show that convincingly, the authors need to show that Ras-Dlg+miR-306/79+dRNF146 clones present the same size as the Ras-Dlg clones. At glance, this does not seem to be the case (compare Fig5G with Figure 5K or M).

After that, the authors state: "Together, these data indicate that miR-306 and miR-79 directly target dRNF146 mRNA, thereby enhancing *JNK* signaling activity and thus exerting the tumor-suppressive effects." The authors make different correlations to reach that conclusion but do not show evidence proving that dRNF146 depletion is in fact enhancing *JNK* activity in those tumors. To conclude that, the authors should prove that *JNK* activity levels are upregulated when dRNF146 is downregulated in those tumors.

To demonstrate their model, the authors should show that Tnks overexpression alone does not affect normal growth in otherwise wild type clones. The authors should also show that miR-309/79 modulation affect Tnks protein levels.

---

## [Author Response]

Essential revisions:1. Show whether 1x or 2x copies of miR-306 or miR-79 cause death in larval or pupal eye discs.

We agree with the comment. Following the suggestion, we have now tested whether 1x or 2x copies of miR-306 or miR-79 cause death in larval eye discs. Our data indicate that, although overexpression of one copy of miR-306 or miR-79 in the eyes had no significant effect on the number of dying cells, overexpression of 2 copies of miR-306 or miR-79 in the eyes significantly increased the number of dying cells. We have now included these new data in Figure 5—figure supplement 4G-M (quantified in Figure 5—figure supplement 4N).

2. Assess whether Rnf146 protein increases in miR 9c/306/79/9b loss-of-function clones in the eye disc.

We do agree with the comment that we should test whether Rnf146 protein increases when miR-306/79 is suppressed. Unfortunately, neither the previous RNF146 antibody (a gift from Hermann Steller) nor a new RNF146 antibody (raised in rabbits against the peptide HSGGGSGEDPAVGSC) worked well in the immunofluorescence assay. Therefore, as an alternative method, we performed Western blots to detect the Rnf146 protein level using the new RNF146 antibody. As shown in Figure 5—figure supplement 3 in the revised manuscript, suppression of miR-306 and miR-79 function by using miRNA sponges indeed increased the endogenous levels of RNF146 protein in the adult eyes.

3. Ensure that the same number of UAS transgenes are present in genotypes that are being compared; this might result in repeating experiments.

We agree with the comment. Following the suggestion, we have now performed a series of new control experiments to make the same number of UAS transgenes present in genotypes that are being compared (please see Figures 1-6) unless it is technically too difficult by *Drosophila* genetics in some experiments (e.g., the experiments in Figure 3—figure supplement 3).

4. Improve Western blots of pJNK by normalizing to the level of total JNK (not α-tubulin) in the sample and increasing sample size to at least three independent replicates. The Western blot data should be displayed as a graph of normalized p-JNK levels with error bars and statistics.

We agree with the comment. Following the suggestion, we have now repeated all the Western blot experiments that used the pJNK antibody and were normalized by the total *JNK* level. All the Western blot data have now been repeated for three times and displayed as a graph with error bars and statistics in the revised Figure 6, Figure 3—figure supplement 2 and Figure 6—figure supplement 1.

5. Improve the Rnf146 Western blot in Figure 5F. At a bare minimum, there should be at least three replicates and quantification in a graph.

We agree with the comment. Following the suggestion, we have now used a new RNF146 antibody (raised in rabbits against the peptide HSGGGSGEDPAVGSC, GenScript antibody service, Nanjing, China). As shown in the revised Figure 5F, we believe that the data quality has now been significantly improved. We repeated this experiment for three times and quantified the data as shown in the revised Figure 5G.

6. Assess whether these miRNAs enhance normally-occurring JNK activity, for example during dorsal closure in the embryo.

We appreciate the valuable comment. Following the suggestion, we have now examined whether expression of these miRNAs enhance normally occurring *JNK* activity during normal development. The pnr-GAL4 fly strain specifically expresses GAL4 in the wing discs in a broad domain corresponding to the central presumptive notum during metamorphosis. Knocking down *Hep*, the *Drosophila JNK* kinase, using the pnr-GAL4 driver generates a split-thorax phenotype caused by reduced *JNK* signaling. On the contrary, overexpression of *Hep* or Eiger, the *Drosophila* TNF that activates *JNK* signaling, using pnr-GAL4 generates a small-scutellum phenotype caused by elevated *JNK* signaling. Similar to the *Hep* or Eiger-induced phanotype, ectopic expression of miR-306 or 79 by the pnr-GAL4 driver resulted in a small-scutellum phenotype, indicating that miR-306/79 enhances normally-occurring *JNK* activity. We have now included these new data as Figure 4—figure supplement 3A-C (quantified in Figure 4—figure supplement 3D) in the revised manuscript.

7. Many controls are not shown. These include but are not limited to:– Figure 1 needs wild-type control (Luc) clones and quantification.– Figure 2 needs wild-type control (Luc) clones and quantification (to the block of panels in Figure 2A-E) and p35-expressing clones and quantification (to the block of panels in Figure 2. F-I).– Figure 3 needs wild-type control clones (Luc alone), and p35-expressing clones the first block of panels (A-I). In the Q-X section, the authors should add Luc alone clones, bsk-DN clones, Ras/dlg clones with quantification. M-P should show control eyes (GMR/+).– Figure 4 A-E and Figure 4-Figure Sup 1, wild-type control clones are required.– Figure 5, Ras-Dlg and dRNF146-OE clones should be included.– Figure 6C, D, should include Tnks alone expressing clones.

We agree with the comment. Following the suggestion, we have now added new controls in the revised Figures (please see Figures 1-6).

8. Show expression of JNK targets – TRE-reporters, pJNK, or puc-lacZ – in miR-306 and 79 clones and in PVR-act, Src64B, Hel25E, and Mahj clones.

We agree with the comment. Following the suggestion, we have now tested whether miR-306 and 79 enhance *JNK* activity in all these tumors or cell competition models using the pJNK antibody. As shown in the revised Figure 4—figure supplement 2D-P, miR-306 or 79 indeed enhanced *JNK* activity in all these tumors or cell competition models.

9. Show that miR-306 and 79 enhance JNK activity in Ras-Dlg tumors by showing the expression of JNK targets as the TRE-reporters, pJNK, or puc-lacZ.

We agree with the comment. Following the suggestion, we have now tested whether miR-306 and 79 enhance *JNK* activity in Ras-Dlg tumors using the pJNK antibody. As shown in the revised Figure 4—figure supplement 2A-C (quantified in 2P), both miR-306 and 79 enhanced *JNK* activity in Ras-Dlg tumors.

10. Show that Ras-Dlg+miR-306/79+dRNF146 clones are the same size as the Ras-Dlg clones. Currently, this does not seem to be the case (compare Fig5G with Figure 5K or M).

We sincerely thank the reviewers and editors for the comment. We indeed noticed that Ras-Dlg+miR-306/79+dRNF146 clones are smaller than Ras-Dlg clones. We have now added a discussion for the possible reason for this phenomenon in the second paragraph of the “Discussion” section in the revised manuscript. Our study identified several putative co-target genes of miR-306 and miR-79 (Figure 5A). Interestingly, some of these genes (Atf3, chinmo, and chn) have been reported to be involved in tumor growth in *Drosophila*. Atf3 encodes an AP-1 transcription factor, which was shown to be a polarity-loss responsive gene acting downstream of the membrane-associated Scrib polarity complex (Donohoe et al., 2018). Knockdown of Atf3 suppresses growth and invasion of Ras^V12^/scrib^-/-^ tumors in the eye-antennal discs (Atkins et al., 2016). Chinmo is a BTB-zinc finger oncogene that is up-regulated by *JNK* signaling in tumors (Doggett et al., 2015). Although loss of chinmo does not significantly suppress tumor growth, overexpression of chinmo with Ras^V12^ or activated Notch is sufficient to promote tumor growth in the eye-antennal discs (Doggett et al., 2015). Chn encodes a zinc finger transcription factor that cooperates with scrib^-/-^ to promote tumor growth (Turkel et al., 2013). Although we found that knockdown of these genes did not activate *JNK* signaling, it is possible that these putative target genes also contribute to the miR-306/miR-79-induced tumor suppression and therefore Ras-Dlg+miR-306/79+dRNF146 clones are smaller than Ras-Dlg clones.

11. Show that JNK activity levels are upregulated when dRNF146 is downregulated in those tumors.

We agree with the comment. Following the suggestion, we have now tested whether *JNK* activity levels are upregulated when dRNF146 is downregulated in Ras-Dlg tumors using the pJNK antibody. As shown in the revised Figure 5—figure supplement 5, knocking down RNF146 indeed enhanced *JNK* activity in Ras-Dlg tumors.

12. Show that Tnks overexpression alone does not affect normal growth in otherwise wild-type clones.

We sincerely thank the reviewers and editors for the comment. Following the suggestion, we have now tested whether Tnks overexpression alone affect normal growth. As shown in Figure 6G-H, Tnks overexpression alone resulted in smaller clone size compared to control. However, considering that overexpression of Tnks alone shows larger clone size than Ras^V12^/dlg^-/-^+Tnks (Figure 6H and J, quantified in Figure 6K), we believe that Tnks suppresses growth of Ras^V12^/dlg^-/-^ tumors by cooperating with *JNK* signaling.

13. Show that miR-309/79 modulation affects Tnks protein levels.

We agree with the comment. Following the suggestion, we have now analyzed the Tnks protein levels when miR-306 or 79 is overexpressed. As shown in Figure 6—figure supplement 1, overexpression of miR-306 or 79 indeed significantly upregulated the Tnks protein level.

Reviewer #1 (Recommendations for the authors):There are several issues that I think should be addressed.Issues with experiments/text:1. The authors should acknowledge the results of Gultekin and Steller in the manuscript and compare results.

We agree with the comment. Actually, we had already described the results of Gultekin and Steller in the first paragraph of the “RNF146 promotes Tnks degradation” part of the “Results” section. However, following the suggestion, we have now added a sentence “A pervious study has indicated that Tnks protein levels were significantly higher in the Rnf146 mutant background than in wild-type (Gultekin and Steller, 2019).” in the second paragraph of the same part.

The main difference between their and our studies is that their study did not clarify whether the up-regulation of Tnks is caused by either elevated Tnks protein synthesis or reduced Tnks protein degradation. In our study, using the cycloheximide (CHX) assay, we clarified that RNF146 promotes the degradation of Tnks protein.

2. The authors state that over-expression of miR-306 or miR-79 (presumably using 1 copy of the transgene) does not reduce growth or WT cells. However, the results in Figure 5, figure supplement 3G-J does not support this model. In these experiments, the authors over-expressed 2 copies of UAS-miR and adult eyes from GMR>2x miR-306 or GMR>2x miR-79 are noticeably smaller than WT and are rough. These results suggest that miR-306 and miR-79 do indeed affect WT cells, which then may make it less likely that miR over-expression can be used as cancer therapy. Can the authors examine death cells in GMR>2x miR-306 or GMR>2x miR-79 larval or pupal discs? Alternatively, what happens when only 1 copy of the UAS-miR is over-expressed?

We agree with the comment. As responded in the Essential Revisions #1, although overexpression of one copy of miR-306 or miR-79 in the eyes had no significant effect on the number of dying cells, overexpression of 2×miR-306 or 2×miR-79 significantly increased the number of dying cells (Figure 5—figure supplement 4G-M, quantified in Figure 5—figure supplement 4N).

3. The authors depleted Rnf146 from GMR>Eiger and saw an enhancement of the eye phenotype. I might be confused, but I thought that it would be better to over-express Rnf146 in GMR>Eiger and look for suppression of the small eye phenotype.

We thank the reviewer for the comment. Our aim was to prove that depletion of Rnf146 have similar effect on the eye phenotype with overexpression of miR-306 or miR-79. Since Rnf146 is a component of the non-canonical *JNK* pathway, a lateral branch but not the major JNKKK-JNKK-*JNK* pathway, overexpression of Rnf146 may not be strong enough to suppress GMR>Eiger induced small-eye phenotype.

4. Most of the experiments are over-expression. Does miR-306 or miR-79 normally regulate levels of Rnf-146 protein? Can you make a clone of the miR 9c/306/79/9b and see Rnf146 protein levels increase?

We agree with the comment. As responded in the Essential Revisions #2, neither the previous RNF146 antibody (a gift from Hermann Steller) nor the new RNF146 antibody (raised in rabbits against the peptide HSGGGSGEDPAVGSC) worked well in the immunofluorescence assay. As an alternative method, we performed Western blots to detect the Rnf146 protein levels using the new RNF146 antibody. As shown in Figure 5—figure supplement 3, suppression of miR-306 and miR-79 function using miRNA sponges indeed promoted the endogenous levels of RNF146 protein in adult eyes.

5. The Rnf146 null allele is adult viable. It might be beyond the scope of this work, but if you made RasV12, dlg-/- clones in an Rnf146 null mutant, the tumors should significantly smaller than in a heterozygous background.

We sincerely thank the reviewer for the comment. However, it turned out that it is so hard to generate Ras^V12^+dlg^-/-^ clones in Rnf146^-/-^ larvae. We have tried to establish flies for several times, but we were not able to succeed in making dlg^m52^, FRT19A; UAS-Ras^V12^; Rnf146^-^/TM6B or TM3 flies.

Issues with figures:1. Figure 1, figure supplement 1, panel AI: the authors need to mention that the dashed horizontal line at 60% comes from the RasV12 dlg-/- clones in Figure 1F.2. Figure 2: Please show Dcp-1 in red or magenta in panels A'-D' as it is very difficult to see white on green.3. Figure 2: panel I – there should be the same number of UAS transgenes in all the genotypes.4. 4. Figure 2, figure supplement 1: Please show Dcp-1 in red or magenta in panels A'-D' as it is very difficult to see white on green.5. Figure 3: panels T and X, there should be the same number of UAS transgenes in all the genotypes.6. Figure 3, panel J: The authors should show a close up of pJNK in WT wing discs with Luc clones as panel J".7. Figure 3, figure supplement 2, panel A: The authors should show a close up of pJNK in WT wing discs with Luc clones as panel A".8. Figure 3, figure supplement 2, panel D: There are several issues with this figure. (1) pJNK levels should be normalize to the level of total JNK in the sample. It is possible that the lower level of pJNK is a result of less JNK protein in GMR>Luc compared to JNK in GMR>miR-306 or GMR-miR-79 and knowing the total level of JNK protein in these cells would allow them to disprove this possibility. Currently, the authors normalize with α-tubulin. (2) The Western blot data should be displayed as a graph of normalized p-JNK levels with error bars and statistics.9. Figure 5, panels B' and C': Please show puc-lacZ in red or magenta in panels A'-D' as it is very difficult to see white on green.10. Figure 5, panel F: The Western blot data should be displayed as a graph of normalized dRNF146 levels with error bars and statistics.11. Figure 5, figure supplement 1, panels A'-F': Please show puc-lacZ in red or magenta as it is very difficult to see white on green.12. The source files are in a.gel format, which I cannot access. Would you please upload them as.tif or.jpeg?13. Include a file with complete genotypes for all figures.

We sincerely thank the reviewer for the comments. Following the suggestions, we have now added new data, revised the figures and descriptions, uploaded the source files for Western blots as.tif, and included files with complete genotypes for each figure as source files in the revised manuscript.

Reviewer #2 (Recommendations for the authors):This is a very complete manuscript. It requires only very few revisions and is appropriate for eLife.1. The authors showed that overexpression of miR-79 and -306 alone was not sufficient to induce a significant phenotype on eye morphology. However, JNK activity is usually not active in eye discs and therefore it cannot be further enhanced. Can expression of these miRNAs enhance normally occurring JNK activity during normal development? One example where the author can address this question is dorsal closure during embryogenesis.

We appreciate the valuable comment. As responded in the Essential Revisions #6, we have now examined whether expression of these miRNAs enhance normally occurring JNK activity during normal development. The pnr-GAL4 driver strain specifically expresses GAL4 in the wing discs in a broad domain corresponding to the central presumptive notum during metamorphosis. Knocking down Hep, the *Drosophila* JNK kinase, using the pnr-GAL4 driver generates a split-thorax phenotype caused by suppressing JNK signaling. On the contrary, ectopic expression of Hep or Eiger using pnr-GAL4 generates a small-scutellum phenotype caused by activation of JNK signaling. Similarly to Hep or Eiger overexpression, ectopic expression of miR-306 or 79 driven by pnr-GAL4 resulted in a small-scutellum phenotype (please see Figure 4—figure supplement 3A-C, quantified in Figure 4—figure supplement 3D in the revised manuscript). These data indicate that miR-306/79 enhances normally-occurring JNK activity.

2. The authors mention that Tankyrase promotes K63-polyubiquitylation of JNK. Is it known how Tankyrase mediates this effect? Even if not, it is worth mentioning this in the text.

We thank the reviewer for the comment. As reported by Ping Li et al., Tnks mediates Poly-ADP ribosylation of JNK, triggers K63-linked poly-ubiquitination of JNK and thereby enhances JNK kinase activity (Li et al., 2018). Following this suggestion, we have now mentioned this in the “RNF146 promotes Tnks degradation” part of the “Results” section in the revised manuscript.

3. The authors did an excellent job in quantifying every experiment shown. However, in the Method section, they did not explain how the expression levels (Dcp-1, pJNK, puc-lacZ, etc.) were measured and quantified.

We thank the reviewer for the comment. The cleaved Dcp-1-positive cell number and pJNK-positive area were calculated using ImageJ and Prism 8 (Graphpad). Following the suggestion, we have now mentioned this in the “Histology” part of the “Materials and Methods” section in the revised manuscript.

Reviewer #3 (Recommendations for the authors):The experiments presented lack some controls. This will allow a more complete comparison between the genetic conditions analyzed.– Figure 1 should show, as panel A, wild type control clones (Luc alone). The quantification of those clones should be shown in panel F.– Figure 2: wild type control clones should be added in the 1st block of results (A-E). p35-expressing clones should be used as controls in the second group of results (F-I).– Figure 3: the first block of results (A-I) should also include wild type control clones (Luc alone), and p35-expressing clones. In the Q-X section, Luc alone clones, bsk-DN clones, Ras/dlg clones should be shown and quantified. M-P should show control eyes (GMR/+).– Figure 4 A-E, and Figure 4-Figure Sup 1, wild type control clones are required.– Figure 5, Ras-Dlg and dRNF146-OE clones should be included.– Figure 6C, D, hould include Tnks alone expressing clones.

We sincerely thank the reviewer for these comments. Following the suggestion, we have now added many controls and revised the figures in the revised manuscript.

To conclude that "miR-306 and miR-79 suppress tumor growth by enhancing JNK signaling", the authors should show that miR-306 and 79 enhance JNK activity in tumors by showing the expression of JNK targets as the TRE-reporters, pJNK, or puc-lacZ.

We agree with the comment. Following the suggestion, as responded in the Essential Revisions #9, we have now tested whether miR-306 and 79 enhance *JNK* activity in Ras-Dlg tumors using the pJNK antibody. As shown in the revised Figure 4—figure supplement 2A-C, both miR-306 and 79 indeed enhanced *JNK* activity in Ras-Dlg tumors.

The title of section "miR-306 and miR-79 enhance JNK signaling in different types of tumors" is misleading and confusing. Among the 4 conditions analyzed – namely, PVR-act, Src64B, Hel25E, and Mahj, only the PVR-act condition (although it lacks the proper control) seems to cause oncogenic growth. As previously mentioned, to claim that those miRNAs enhance JNK activity, the authors should show the expression of JNK targets as the TRE-reporters, pJNK, or puc-lacZ.

We appreciate the valuable comment. We have now revised the title to “miR-306 and miR-79 enhance *JNK* signaling stimulated by different upstream signaling”. Following the suggestion, as responded in the Essential Revisions #8, we have now tested whether miR-306 and 79 enhance *JNK* activity in all these tumors or cell competition models using the pJNK antibody. As shown in the revised Figure 4—figure supplement 2D-O, miR-306 or 79 indeed enhanced *JNK* activity in all these tumors or cell competition models.

Figure 4 K-O. The authors show that "non-autonomous overgrowth of surrounding wild-type tissue by Src64B-overexpressing clones". Is this observation relevant for this manuscript? It seems to come out of the blue and I can´t find the connection with the rest of results presented here.

The non-autonomous overgrowth of surrounding wild-type tissue is caused by Src64B-overexpressing cells (Enomoto and Igaki, 2013). Coexpression of miR-9c/306/79/9b cluster, miR-306, or miR-79 kills Src64B-overexpressing cells in the eye-antennal discs (Figure 4H-L) and thus suppresses the non-autonomous overgrowth of surrounding wild-type tissue (Figure 4M-Q). This data indicated that miR-306 or miR-79 not only suppresses tumor growth caused by autonomous overgrowth (e.g., Ras^V12^/dlg^-/-^, Ras^V12^/lgl^-/-^ and PVR^act^), but also suppresses tumor growth caused by non-autonomous overgrowth.

Related to the regulation of dRNF146 by miRNA-305/79, the results shown in D and E show that the predicted binding site is sensitive to the levels of those miRNAs. However, results showing that the proposed regulation is indeed taking place in vivo and is physiologically relevant are missing. The western presented in Figure 5F is very poor. Results with better quality need to be provided to show that convincingly. In line with that, showing the protein levels in miR-306 and 79-expressing clones in the eye disc will provide a convincing piece of data.

We agree with the comment. As responded in the Essential Revisions #2 and #5, we have now used a new RNF146 antibody (raised in rabbits against the peptide HSGGGSGEDPAVGSC, GenScript antibody service, Nanjing, China). As shown in the revised Figure 5F, we believe the data quality have now significantly improved. We repeated this experiment for three times and quantified the data as shown in the revised Figure 5G. Unfortunately, neither the previous RNF146 antibody (a gift from Hermann Steller) nor the new RNF146 antibody worked well in the immunofluorescence assay. As an alternative method, we performed Western blots to detect the Rnf146 protein levels using the new RNF146 antibody. As shown in Figure 5—figure supplement 3 in the revised manuscript, suppression of miR-306 and miR-79 functions using miRNA sponges indeed promoted the endogenous levels of RNF146 protein in the adult eyes.

The authors claim: "Furthermore, overexpression of dRNF146 cancelled the tumor-suppressive effect of miR-306 or miR-79 on RasV12/dlg-/- tumors (Figure 5J-M, quantified in Figure 5N)." To show that convincingly, the authors need to show that Ras-Dlg+miR-306/79+dRNF146 clones present the same size as the Ras-Dlg clones. At glance, this does not seem to be the case (compare Fig5G with Figure 5K or M).

We thank the reviewer for the comment. We have now replaced the word “cancelled” with “weakened” in the revised manuscript. As responded in the Essential Revisions #10, we indeed noticed that Ras-Dlg+miR-306/79+dRNF146 clones are smaller than Ras-Dlg clones. We have now added a discussion for the possible reason for this phenomenon in the second paragraph of the “Discussion” section in the revised manuscript. Our study identified several putative co-target genes of miR-306 and miR-79 (Figure 5A). Interestingly, some of these genes (Atf3, chinmo, and chn) have been reported to be involved in tumor growth in *Drosophila*. Atf3 encodes an AP-1 transcription factor which was shown to be a polarity-loss responsive gene acting downstream of the membrane-associated Scrib polarity complex (Donohoe et al., 2018). Knockdown of Atf3 suppresses growth and invasion of Ras^V12^/scrib^-/-^ tumors in the eye-antennal discs (Atkins et al., 2016). Chinmo is a BTB-zinc finger oncogene that is up-regulated by *JNK* signaling in tumors (Doggett et al., 2015). Although loss of chinmo does not significantly suppress tumor growth, overexpression of chinmo with Ras^V12^ or activated Notch is sufficient to promote tumor growth in eye-antennal discs (Doggett et al., 2015). Chn encodes a zinc finger transcription factor that cooperates with scrib^-/-^ to promote tumor growth (Turkel et al., 2013). Although we found that knockdown of these genes did not activate *JNK* signaling, it is possible that these putative target genes also contribute to the miR-306/miR-79-induced tumor suppression and thus Ras-Dlg+miR-306/79+dRNF146 clones are smaller than Ras-Dlg clones..

After that, the authors state: "Together, these data indicate that miR-306 and miR-79 directly target dRNF146 mRNA, thereby enhancing JNK signaling activity and thus exerting the tumor-suppressive effects." The authors make different correlations to reach that conclusion but do not show evidence proving that dRNF146 depletion is in fact enhancing JNK activity in those tumors. To conclude that, the authors should prove that JNK activity levels are upregulated when dRNF146 is downregulated in those tumors.

We agree with the comment. As responded in the Essential Revisions #11, we have now tested whether *JNK* activity levels are upregulated when dRNF146 is downregulated in Ras-Dlg tumors using the pJNK antibody. As shown in the revised Figure 5—figure supplement 5, knocking down RNF146 indeed enhanced *JNK* activity in Ras-Dlg tumors.

To demonstrate their model, the authors should show that Tnks overexpression alone does not affect normal growth in otherwise wild type clones. The authors should also show that miR-309/79 modulation affect Tnks protein levels.

We sincerely thank the reviewer for the comment. As responded in the Essential Revisions #12 and #13, we have now tested whether Tnks overexpression alone affect normal growth. As shown in Figure 6G-H, Tnks overexpression alone resulted in smaller clone size compared to control. However, considering that overexpression of Tnks alone shows larger clone size than Ras^V12^/dlg^-/-^+Tnks (Figure 6H and J, quantified in Figure 6K), we believe that Tnks suppresses growth of Ras^V12^/dlg^-/-^ tumors by cooperating with *JNK* signaling. We also detected the Tnks protein levels when miR-306 or 79 is overexpressed. As shown in Figure 6—figure supplement 1, overexpression of miR-306 or 79 significantly upregulated Tnks protein level.